# Sorting of secretory proteins at the *trans-Golgi network* by human TGN46

Pablo Lujan[1], Carla Garcia-Cabau[2†], Yuichi Wakana[3†], Javier Vera Lillo[1], Carmen Rodilla-Ramírez[1], Hideaki Sugiura[3], Vivek Malhotra[4,5,6], Xavier Salvatella[2,6], Maria F Garcia-Parajo[1,6], Felix Campelo[1*]

[1]ICFO-Institut de Ciencies Fotoniques, The Barcelona Institute of Science and Technology, Barcelona, Spain; [2]Institute for Research in Biomedicine (IRB Barcelona), The Barcelona Institute of Science and Technology, Barcelona, Spain; [3]School of Life Sciences, Tokyo University of Pharmacy and Life Sciences, Tokyo, Japan; [4]Centre for Genomic Regulation, The Barcelona Institute of Science and Technology, Barcelona, Spain; [5]Universitat Pompeu Fabra, Barcelona, Spain; [6]Institució Catalana de Recerca i Estudis Avançats (ICREA), Barcelona, Spain

**\*For correspondence:**
felix.campelo@icfo.eu

[†]These authors contributed equally to this work

**Abstract** Secretory proteins are sorted at the *trans*-Golgi network (TGN) for export into specific transport carriers. However, the molecular players involved in this fundamental process remain largely elusive. Here, we identified the human transmembrane protein TGN46 as a receptor for the export of secretory cargo protein PAUF in CARTS – a class of protein kinase D-dependent TGN-to-plasma membrane carriers. We show that TGN46 is necessary for cargo sorting and loading into nascent carriers at the TGN. By combining quantitative fluorescence microscopy and mutagenesis approaches, we further discovered that the lumenal domain of TGN46 encodes for its cargo sorting function. In summary, our results define a cellular function of TGN46 in sorting secretory proteins for export from the TGN.

## eLife assessment

This study provides the **fundamental** insight that TGN46, a single-pass membrane protein, acts as a cargo receptor for proteins at the Trans-Golgi Network. The authors demonstrate that the luminal domain of TGN46 is crucial for the incorporation of the soluble secretory protein PAUF into CARTS, a class of vesicles mediating TGN to surface traffic. The data presented are **compelling**, yielding a clear model for the sorting of cargos destined for secretion.

## Introduction

Secretory proteins synthesized into the lumen of the ER are transported to the Golgi apparatus, where they are further processed and eventually sorted at the *trans*-Golgi network (TGN) for delivery to their respective destinations (*Stalder and Gershlick, 2020*; *Wakana and Campelo, 2021*; *Spang, 2015*; *von Blume and Hausser, 2019*; *Di Martino et al., 2019*). The accurate and timely export of cargo proteins – such as collagens, hormones, or neurotransmitters – is essential for the maintenance of cell and tissue homeostasis. Accordingly, errors in this process can have severe physiological and patho-physiological consequences (*Zappa et al., 2018*). Despite the importance of this process, the molecular machinery and the mechanisms of cargo sorting at the TGN remain largely elusive (*Guo et al., 2014*; *Boncompain and Weigel, 2018*; *Ford et al., 2021*; *Ramazanov et al., 2021*). In mammalian cells, lysosomal hydrolases bearing a mannose 6-phosphate are sorted by the mannose 6-phosphate receptors into clathrin-coated vesicles for their transport to the endo-lysosomal system (*Kienzle and*

*von Blume, 2014*; *Braulke and Bonifacino, 2009*). In contrast, no protein coats have been identified for most export routes from the TGN – especially for those destined to the cell surface (*Stalder and Gershlick, 2020*; *Wakana and Campelo, 2021*) – which has hampered the identification of cargo receptors or of other sorting modules. The discovery by Malhotra et al. of a $Ca^{2+}$-dependent mechanism for the sorting of a subset of soluble secretory cargoes is beginning to reveal the intricacies of this supposedly receptor-independent process (*von Blume et al., 2009*; *Curwin et al., 2012*). This mechanism relies on the coordinated action of various cellular components, including the actin filament-severing protein cofilin (*von Blume et al., 2009*; *Curwin et al., 2012*), the $Ca^{2+}$ pump SPCA1 (*von Blume et al., 2011*; *Kienzle et al., 2014*), the oligomeric lumenal $Ca^{2+}$-binding protein Cab45 (*Crevenna et al., 2016*; *von Blume et al., 2012*), and the sphingolipid sphingomyelin (SM) (*Deng et al., 2016*; *Deng et al., 2018*). The export from the Golgi by this pathway is independent of any coat proteins identified thus far. However, how other soluble cargo proteins are sorted for their export from the TGN remains unknown.

The single-pass type-I transmembrane (TM) protein TGN46 – or TGN38 in rodents (*Ponnambalam et al., 1996*) – has been proposed to function as a secretory cargo receptor (*Stanley and Howell, 1993*; *Jones et al., 1993*; *Wang et al., 1995*; *Banting and Ponnambalam, 1997*; *McNamara et al., 2004*; *Wang and Howell, 2000*; *Ramazanov et al., 2021*). However, direct evidence for a role of TGN46 in Golgi export is lacking. TGN38 was first identified in rat cells and shown to have a well-defined steady-state localization at the TGN (*Luzio et al., 1990*). As a result, TGN38 and its orthologs have become extensively employed as TGN marker proteins. The amino acid sequence of rat TGN38 is largely conserved among other species, including humans (>80% amino acid identity between rat TGN38 and human TGN46), with highly conserved cytosolic and TM domains and a somewhat less conserved lumenal domain (>60% overall identity but only ~25% in the bulk of the lumenal domain) (*Ponnambalam et al., 1996*). Despite having a steady-state TGN localization, TGN38/TGN46 rapidly cycles between the TGN and the plasma membrane (PM), exiting the TGN in specific carriers called CARTS (carriers of the TGN to the cell surface) (*Wakana et al., 2012*) and recycling from the PM back to the Golgi via early endosomes (*Banting and Ponnambalam, 1997*; *Bos et al., 1993*; *Rajasekaran et al., 1994*; *Mallet and Maxfield, 1999*; *Humphrey et al., 1993*). CARTS were first identified by immuno-isolation of TGN46-containing transport carriers from the TGN to the PM (*Wakana et al., 2012*). CARTS are formed in a protein kinase D (PKD)-dependent manner and contain the Rab6/Rab8 GTPases. CARTS traffic specific secretory cargoes, such as pancreatic adenocarcinoma upregulated factor (PAUF) and lysozyme C (LyzC), while exclude others, such as collagen I or vesicular stomatitis virus G-protein (VSVG) (*Wakana et al., 2012*; *Wakana et al., 2013*; *Wakana and Campelo, 2021*; *Lujan et al., 2021*). However, how secretory cargoes are sorted and specifically packed into CARTS is yet to be understood. Furthermore, whether TGN46 plays an active role in CARTS biogenesis or it is just a passive TM cargo of this pathway is still unknown.

Here, we demonstrate that TGN46 plays a key role in constitutive protein secretion, specifically in the sorting of the soluble cargo protein PAUF into CARTS for TGN export. Our data reveal that the topological determinants that describe the proper intracellular and intra-Golgi localization of TGN46, as well as its own incorporation in CARTS, are mainly contained in its lumenal domain. Notably, we find that the lumenal domain of TGN46 is both necessary and sufficient for the export of cargo proteins into CARTS and, moreover, that this domain mediates the cargo sorting function of TGN46. Altogether, our data suggest an essential role for TGN46 in the sorting of cargo proteins into transport carriers (CARTS) at the TGN.

## Results

### TGN46 is required for cargo sorting into CARTS

As TGN46 has been suggested, but not shown, to function in cargo sorting at the TGN, we decided to test this hypothesis. We generated a TGN46-lacking line of HeLa cells by CRISPR/Cas9-mediated knock out of *TGOLN2* – the gene encoding for human TGN46 (TGN46-KO cell line) (*Figure 1A* and *Figure 1—figure supplement 1A*). To test if TGN46 is required for CARTS-mediated secretion, we transiently expressed a CARTS-specific soluble cargo protein, PAUF-MychIs, in both parental and TGN46-KO cells. Both the medium and cell lysates were collected and analyzed by Western blotting with an anti-Myc antibody to detect the secreted and internal PAUF-MychIs pools, respectively.

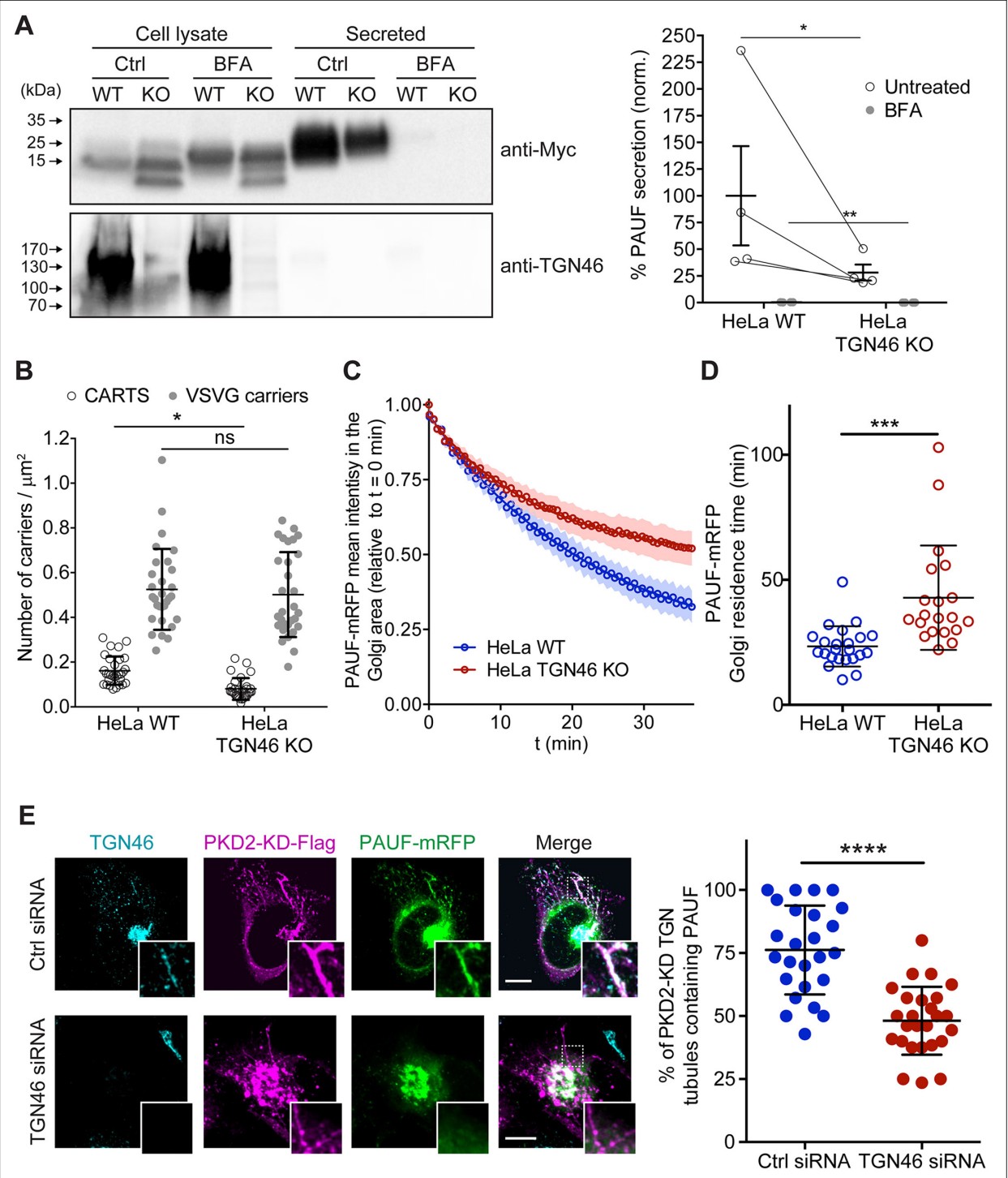

**Figure 1.** TGN46 is required for cargo sorting and loading into CARTS. (**A**) Pancreatic adenocarcinoma upregulated factor (PAUF) secretion assay in parental (WT) and CRISPR/Cas9-mediated TGN46 knockout (KO) HeLa cells. Cells expressing PAUF-MychHis were incubated for 4 hr in fresh medium without serum in the absence (Ctrl) or presence of Brefeldin A (BFA), after which the cells were lysed. Equal amounts of cell lysates and their corresponding secreted fraction (medium) were analyzed by Western blotting (left) using an anti-Myc antibody (top blot) and anti-TGN46 antibody (bottom blot). BFA was included as a positive control of secretion inhibition. Quantification (right) of the ratio of secreted/internal PAUF signal normalized (norm.) as a percentage to the average value of the WT, Ctrl condition. $n$ = 4 independent experiments. Individual values shown with connecting lines between paired datasets (shown only for untreated condition, for clarity), with mean ± stdev. Paired $t$ test (*p ≤ 0.05; **p ≤ 0.01). (**B**) Number of carriers (CARTS, empty circles; VSVG carriers, gray circles) per unit area observed in WT or TGN46-KO HeLa cells expressing either PAUF-mRFP (CARTS marker) and VSVG-HA (VSVG carrier marker). At least 10 cells from each of $n$ = 3 independent experiments were quantified. Individual

*Figure 1 continued on next page*

*Figure 1 continued*

values shown, with mean ± stdev. Unpaired two-tailed *t* test (ns, p > 0.05; *p ≤ 0.05). (**C**) Relative fluorescence intensity average time trace (mean ± standard error of the mean [s.e.m.]) of fluorescence loss in photobleaching (FLIP) experiments performed in WT or TGN46-KO HeLa cells expressing PAUF-mRFP. (**D**) Quantification of the PAUF-mRFP Golgi residence time as obtained from the FLIP experiments, as the one shown in (**C**). Between 7 and 12 cells from each of *n* = 3 independent experiments were quantified. Individual values shown, with mean ± stdev. Unpaired two-tailed *t* test (***p ≤ 0.001). (**E**) Left panels are confocal fluorescence microscopy images of HeLa cells transfected with control (Ctrl) or TGN46 siRNA, which were also transfected with PKD2-KD-Flag and PAUF-mRFP. Cells were fixed at steady state and labeled with anti-TGN46 (cyan) and anti-Flag (magenta) antibodies. PAUF-mRFP fluorescence signal is shown in green. Scale bars are 10 μm, magnifications of the boxed regions are shown. In the right panel, the quantification of the percentage of PKD2-KD-induced tubules that containing PAUF-mRFP in control (Ctrl) or TGN46 siRNA-treated cells. At least 10 cells from each of *n* = 3 independent experiments. Individual values shown, with mean ± stdev. Unpaired two-tailed *t* test (****p ≤ 0.0001).

The online version of this article includes the following source data and figure supplement(s) for figure 1:

**Source data 1.** Uncropped images of the membranes of the Western blotting shown in *Figure 1A*.

**Figure supplement 1.** TGN46 is required for cargo sorting and loading into CARTS.

In TGN46-KO cells, PAUF-MycHis secretion was reduced by ~75% as compared to control cells (*Figure 1A*). We noticed the appearance of a lower molecular weight form of PAUF-MycHis in the lysate of TGN46-KO cells (*Figure 1A*), suggesting that the processing (e.g., glycosylation) of intra-cellularly accumulated PAUF-MycHis is incomplete, and opening the possibility of a role for TGN46 in facilitating cargo glycosylation. Immunofluorescence microscopy of cells expressing PAUF-mRFP (monomeric red fluorescent protein) revealed its presence in a perinuclear compartment in both parental and TGN46-KO cell lines, indicating that PAUF-mRFP efficiently exits the ER in the absence of TGN46 (*Figure 1—figure supplement 1A*).

To test whether TGN46 is specifically involved in the secretion of CARTS-mediated cargoes or rather plays a general role in TGN export, we simultaneously monitored Golgi export of PAUF-mRFP (a CARTS cargo) and VSVG-HA (a non-CARTS cargo) (*Wakana et al., 2012*; *Wakana and Campelo, 2021*) in parental and TGN46-KO cells. We used the standard temperature-induced block in cargo export at the TGN to monitor synchronous exit of VSV-G or PAUF in control and TGN46-lacking cells. Cells were first incubated at 20°C for 2 hr to block TGN export, and subsequently shifted to 37°C for 15 min to allow for synchronized export from the TGN, after which cells were fixed and the number of cargo-containing vesicular structures per unit area was analyzed by confocal fluorescence micros-copy (*Figure 1B* and *Figure 1—figure supplement 1B*). Our data show that the number density of PAUF-mRFP-positive punctate structures – which define CARTS (*Wakana et al., 2012*) – is signifi-cantly reduced by ~50% in TGN46-KO cells as compared to parental cells (*Figure 1B*). These results agree with and complement the observed inhibition of PAUF secretion in the absence of TGN46 (*Figure 1A*). In contrast, the number density of VSVG-containing carriers was unaltered upon TGN46 depletion (*Figure 1B* and *Figure 1—figure supplement 1B*), indicating that TGN46 selectively func-tions in the export of a subset of cargo proteins at the TGN. In addition, TGN46 depletion did not increase the proportion of CARTS (defined as carriers positive for PAUF) that also contained VSVG (*Figure 1—figure supplement 1C*). These data strongly suggest that, in the absence of TGN46, PAUF is not re-routed toward alternative TGN export routes (particularly to that of VSV-G-containing trans-port carriers), and therefore bona fide CARTS are still formed, although at a strongly reduced rate. Hence, we will use PAUF as a marker protein for CARTS throughout this article.

We next assessed by fluorescence loss in photobleaching (FLIP) microscopy the residence time of PAUF in the Golgi apparatus, which is a dynamic measure of the Golgi export rate of this cargo (*Figure 1—figure supplement 1D*). In our FLIP experiments, we systematically photobleached by using a high intensity laser the entire pool of fluorescent protein (e.g., PAUF-mRFP) located outside of the perinuclear Golgi area, and measured the decay of the fluorescence intensity of the Golgi area as function of time (*Figure 1—figure supplement 1D*). Thus, for cargo proteins that are rapidly exported from the Golgi, the fluorescence intensity in the Golgi area is expected to rapidly decay, whereas for resident Golgi proteins, the fluorescence should decay more slowly. A variant of FLIP microscopy, known as inverse fluorescence recovery after photobleaching (iFRAP) microscopy, has been previously used to monitor the dynamics of Golgi export (*Hirschberg et al., 1998*; *Patterson et al., 2008*). The main distinction between iFRAP and FLIP lies in the frequency of photobleaching. While in iFRAP, photobleaching is performed only once at the beginning of the experiment, FLIP involves repeated photobleaching (FLIP is sometimes also referred to as 'repeated iFRAP') (*Ishikawa-Ankerhold et al.,*

*2012*). Our FLIP results showed that PAUF-mRFP has a longer Golgi residence time in TGN46-KO cells (~40 min) as compared to control cells (~25 min), and therefore TGN46 is necessary for an efficient and fast export of PAUF (*Figure 1C, D* and *Figure 1—figure supplement 1D*). Taken together, our results indicate that TGN46 is necessary for CARTS-mediated PAUF export from the TGN.

## TGN46 is required for cargo loading into CARTS

Cargo-loaded CARTS detach from the TGN once the neck of the nascent carrier is severed through a membrane fission reaction, a process regulated by the kinase activity of PKD (*Wakana et al., 2012*). Accordingly, expression of a kinase-dead mutant of PKD (PKD-KD) leads to the formation of cargo-loaded tubules (fission-defective CARTS precursors) at the TGN (*Malhotra and Campelo, 2011*; *Wakana and Campelo, 2021*). We took advantage of this to test if TGN46 is needed for cargo loading into CARTS and/or for membrane bending. We co-transfected PAUF-mRFP and PKD2-KD-Flag in control and TGN46-depleted HeLa cells, after which cells were fixed and processed for immunofluorescence microscopy. We first observed that TGN46 depletion did not affect the number or appearance of PKD2-KD-Flag-positive TGN tubules per cell (*Figure 1E* and *Figure 1—figure supplement 1E*), suggesting that TGN46 is not necessary for membrane bending. Next, we observed that in control (TGN46-positive) cells, a large fraction of Golgi tubules contained both TGN46 and PAUF-mRFP (~75 ± 15% of tubules containing PAUF) (*Figure 1E*), as previously described (*Liljedahl et al., 2001*; *Wakana et al., 2012*). In contrast, and remarkably, PAUF-mRFP was absent from the majority of the tubules induced by PKD2-KD-Flag in TGN46-depleted cells (~45 ± 10% of tubules containing PAUF) (*Figure 1E* and *Figure 1—figure supplement 1E*). These findings strongly indicate that TGN46 is required for sorting and loading the secretory cargo protein PAUF into nascent CARTS at the TGN.

## TGN46 export in CARTS is not dependent on TGN46 cytosolic tail signals

Although a number of sequence motifs have been reported to serve as retention signals for Golgi-resident proteins (*Welch and Munro, 2019*; *Lujan and Campelo, 2021*), less is known about how cargoes are selectively sorted into transport carriers budding from the TGN. Particularly, sorting mechanisms for CARTS-specific cargoes remain elusive (*Wakana and Campelo, 2021*). To investigate how TGN46 sorts cargoes into CARTS, we employed a mutagenesis approach to examine whether TGN46 possesses specific topological determinants (such as domains, regions, or motifs) responsible for its cargo sorting and packaging function. As TGN46 is a single-pass type-I TM protein of 437 amino acids (human TGN46, UniProt identifier O43493-2), we investigated whether functional cargo-packaging modules can be found in (1) the N-terminal lumenal domain (amino acids 22–381), (2) the single TM domain (TMD; amino acids 382–402), and/or (3) the short C-terminal cytosolic tail (amino acids 403–437). In particular, we generated a battery of mutant proteins and evaluated how they behave in terms of (1) their fine intra-TGN protein localization, (2) their specific sorting into CARTS, and (3) their Golgi export rate.

We start by investigating the existence of sorting signals in the cytosolic tail of TGN46. This tail region contains a tyrosine-based cytosolic motif (YXXΦ; where X represents any amino acid, and Φ is an amino acid with bulky hydrophobic side chains *Ohno et al., 1996*; in TGN46: YQRL, amino acids 430–433). This motif acts as an endocytic signal for PM internalization and trafficking back to the TGN, and is therefore crucial for the steady-state localization of TGN46 at TGN (*Banting and Ponnambalam, 1997*; *Stanley and Howell, 1993*; *Rajasekaran et al., 1994*; *Bos et al., 1993*; *Humphrey et al., 1993*; *Banting et al., 1998*). We expressed in HeLa cells a GFP-tagged deletion mutant of TGN46 lacking the cytosolic domain but maintaining the lumenal domain and TMD (GFP-TGN46-Δcyt) (*Figure 2A*) and analyzed its intracellular localization by fluorescence microscopy. As previously reported (*Bos et al., 1993*; *Humphrey et al., 1993*), GFP-TGN46-Δcyt was partially present at the *trans*-Golgi membranes/TGN and partially at the PM, as qualitatively seen in the micrographs (*Figure 2*). We and others have previously shown that the *trans*-Golgi/TGN membranes are laterally compartmentalized into export domains (positive for TGN46) and processing domains (positive for the late-Golgi glycosylation enzyme sialyltransferase, ST) (*van Galen et al., 2014*; *Patterson et al., 2008*; *Tie et al., 2022*). We therefore quantitatively measured whether GFP-TGN46-Δcyt partitions into export or processing subdomains. To that end, we co-expressed either GFP-TGN46 WT or GFP-TGN46-Δcyt together with TGN46-mRFP or ST-mCherry, and measured the degree of perinuclear

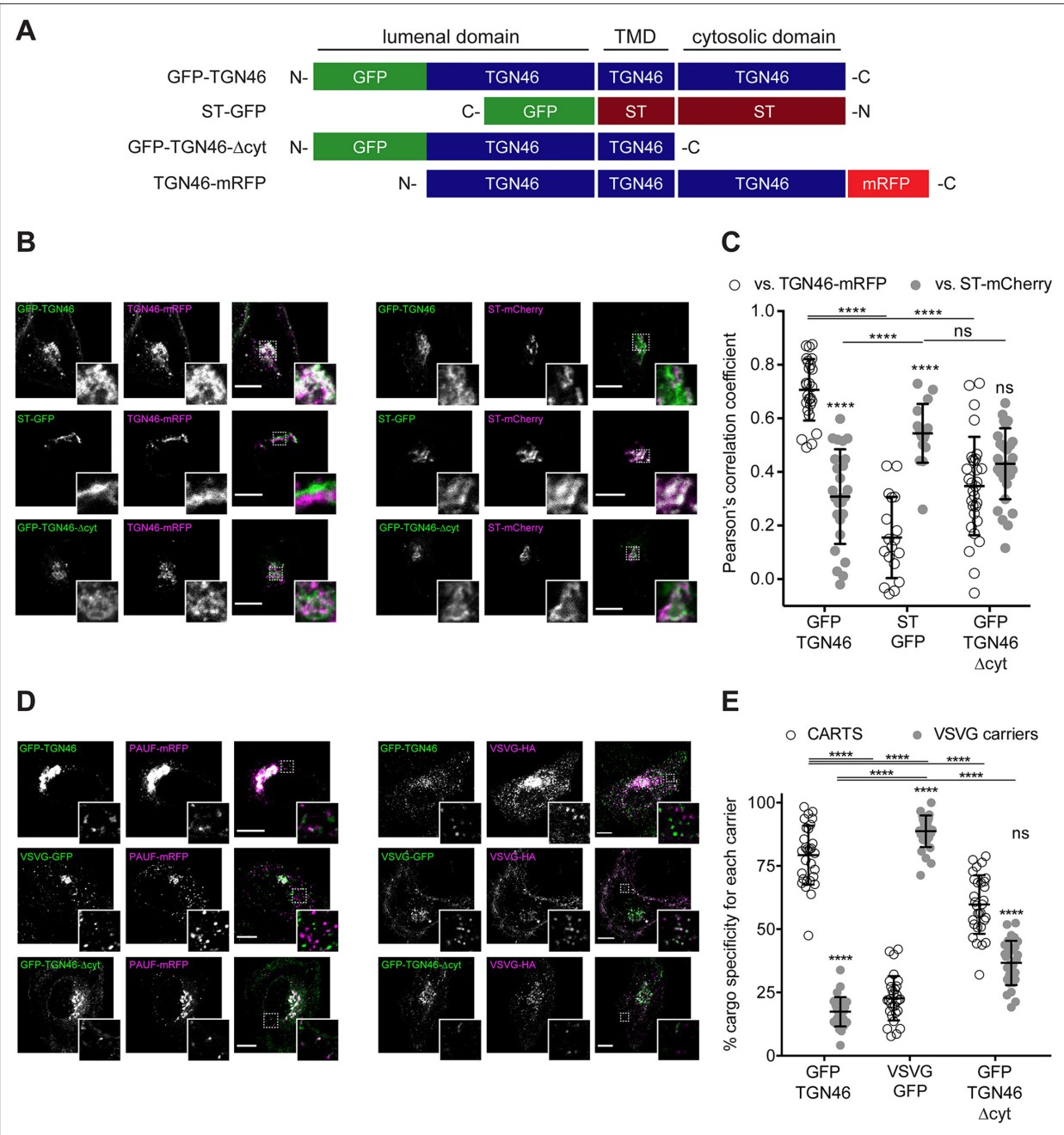

**Figure 2.** TGN46 export in CARTS is not dependent on cytosolic tail signals. (**A**) Schematic representation of construct domain topology. Notice that type-I proteins (e.g., GFP-TGN46) have a lumenal N-terminal domain, whereas type-II proteins (e.g., ST-GFP) have a cytosolic N-terminal domain. TMD: transmembrane domain. (**B**) HeLa cells co-expressing the different indicated proteins (green and magenta channels) were fixed, and the localization of those proteins was monitored by fluorescence confocal microscopy. Insets correspond to zoom-in areas of the dashed, white boxed areas. (**C**) Pearson's correlation coefficient between the perinuclear fluorescence signal of the x-axis indicated proteins with respect to TGN46-mRFP (empty circles) or ST-mCherry (gray circles), measured from confocal micrographs in (**B**). Results are from at least 10 cells from each of n = 3 independent experiments (individual values shown, with mean ± stdev; ns, p > 0.05; ****p ≤ 0.0001). (**D**) HeLa cells co-expressing the different indicated proteins (green and magenta channels) were fixed, processed for immunostaining when required, and the localization of those proteins was monitored by fluorescence confocal microscopy. Insets correspond to zoom-in areas of the dashed, white boxed areas. (**E**) Percentage of transport carriers containing each of the cargoes described on the x-axis that are also positive for pancreatic adenocarcinoma upregulated factor (PAUF; CARTS, empty circles) or VSVG (VSVG carriers, gray circles), as measured from confocal micrographs in (**D**). Results are from at least 10 cells from each of n = 3 independent experiments (individual values shown, with mean ± stdev; ns, p > 0.05; ****p ≤ 0.0001). Scale bars in (**B, D**) are 10 μm.

colocalization between the different protein pairs (Pearson's correlation coefficient, *Dunn et al., 2011*; *Figure 2B, C*). Our data show that GFP-TGN46-Δcyt has a lower degree of intra-Golgi colocalization with respect to TGN46-mRFP and a higher one with respect to ST-mCherry, as compared to its WT counterpart (*Figure 2B, C*). Hence, GFP-TGN46-Δcyt shows an intra-Golgi localization pattern more closely resembling that of ST-GFP-positive processing domains (*Figure 2B, C*). These data indicate that the cytosolic domain of TGN46 confers an overall TGN localization by contributing not only to the recycling from the PM, as previously reported (*Bos et al., 1993*; *Humphrey et al., 1993*), but also to its fine steady-state partitioning into TGN export domains and exclusion from processing domains. We next tested how GFP-TGN46-Δcyt is exported from the TGN. We first monitored the incorporation of GFP-TGN46-Δcyt into CARTS. HeLa cells co-overexpressing GFP-TGN46-Δcyt together with either PAUF-mRFP (CARTS marker) or VSVG-HA (a CARTS-independent TM cargo protein) were incubated at 20°C for 2 hr to inhibit TGN export, shifted to 37°C for 15 min to re-initiate cargo export, fixed, and imaged by fluorescence microscopy. We then analyzed the fraction of cargo-containing carriers (for three different cargoes: GFP-TGN46 WT, VSVG-HA, and GFP-TGN46-Δcyt) that were also positive for PAUF-mRFP (CARTS) or VSVG-HA (VSVG carriers). Our results indicated that GFP-TGN46-Δcyt is preferentially exported in CARTS and not in VSVG carriers, although this mutant cargo protein showed a modest but statistically significant reduction in its CARTS specificity as compared to GFP-TGN46 WT (*Figure 2D, E*). We next performed FLIP experiments to measure the Golgi residence time of GFP-TGN46-Δcyt, and compare it with that of GFP-TGN46 WT. It is important to emphasize that, owing to the execution of FLIP experiments involving repeated photobleaching of the pool of fluorescent proteins outside the Golgi area, we were able to monitor the actual protein export rate (*Figure 1—figure supplement 1D*). This experimental design minimizes the impact of TGN46 recycling back to the TGN, as would occur in iFRAP experiments involving a single photobleaching step. Interestingly, GFP-TGN46-Δcyt exits the Golgi at a rate that is indistinguishable from that of GFP-TGN46 WT (*Figure 3A, B* and *Figure 3—figure supplement 1A*). Taken together, our results suggest that, although the cytosolic tail of TGN46 is important for PM recycling, it does not play a major role in mediating TGN46 incorporation into CARTS for export out of the TGN.

## TGN46 intra-Golgi localization and CARTS specificity are insensitive to the length and composition of its TMD

We next investigated if the information encoded in the amino acid sequence of the TGN46 TMD determines its incorporation into CARTS. The hydrophobic matching hypothesis is a TMD-based sorting mechanism that has been proposed to contribute to the retention of Golgi-resident TM proteins (*Welch and Munro, 2019*; *Lujan and Campelo, 2021*). This mechanism is based on the hypothesis that single-pass TM proteins preferably partition into membrane regions of a thickness that best matches the hydrophobic length of the protein TMD (*Munro, 1991*; *Sharpe et al., 2010*; *Munro, 1995*). The membrane thickness increases along the secretory pathway (*Mitra et al., 2004*), which correlates with the average TMD length of the proteins of these organelles (*Sharpe et al., 2010*). Hence, according to the TMD-based retention hypothesis, the separation between Golgi-resident proteins and cargoes destined for export from the TGN is attributed to the distinct physical properties of these domains (*Sharpe et al., 2010*; *Munro, 1995*; *Quiroga et al., 2013*; *Cosson et al., 2013*). TGN46 is a type-I protein with a single TMD of 21 amino acids, whereas ST (ST6Gal-I) is a type-II protein with a shorter TMD of 17 amino acids (*Figure 4A*). Thus, we generated two different TGN46 mutants by altering its TMD sequence to test whether the length and/or amino acid composition of the TMD of TGN46 is important for its characteristic intra-Golgi localization, selective CARTS-mediated export, and Golgi residence time. The first mutant (TGN46-shortTMD) is a TGN46 deletion that lacks four amino acids (AAIL) in the central region of the TMD. For the second mutant (TGN46-ST TMD), we replaced the TGN46 TMD by that of ST, while keeping the topology and, therefore, the correct TMD orientation with respect to the bilayer (*Figure 4A*). As compared to TGN46 WT, both mutants have shorter TMDs (17 amino acids) and while TGN46-shortTMD maintains the overall amino acid composition, TGN46-ST TMD has a completely altered TMD composition (*Figure 4A*).

First, we expressed different combinations of these proteins in HeLa cells, and monitored their intracellular localization by fluorescence microscopy. Altering the TMD of TGN46 had no discernible impact on intra-Golgi localization (*Figure 4B, C*), suggesting that the hydrophobic matching mechanism is not a major determinant for the fine intra-Golgi localization of TGN46. Interestingly, even

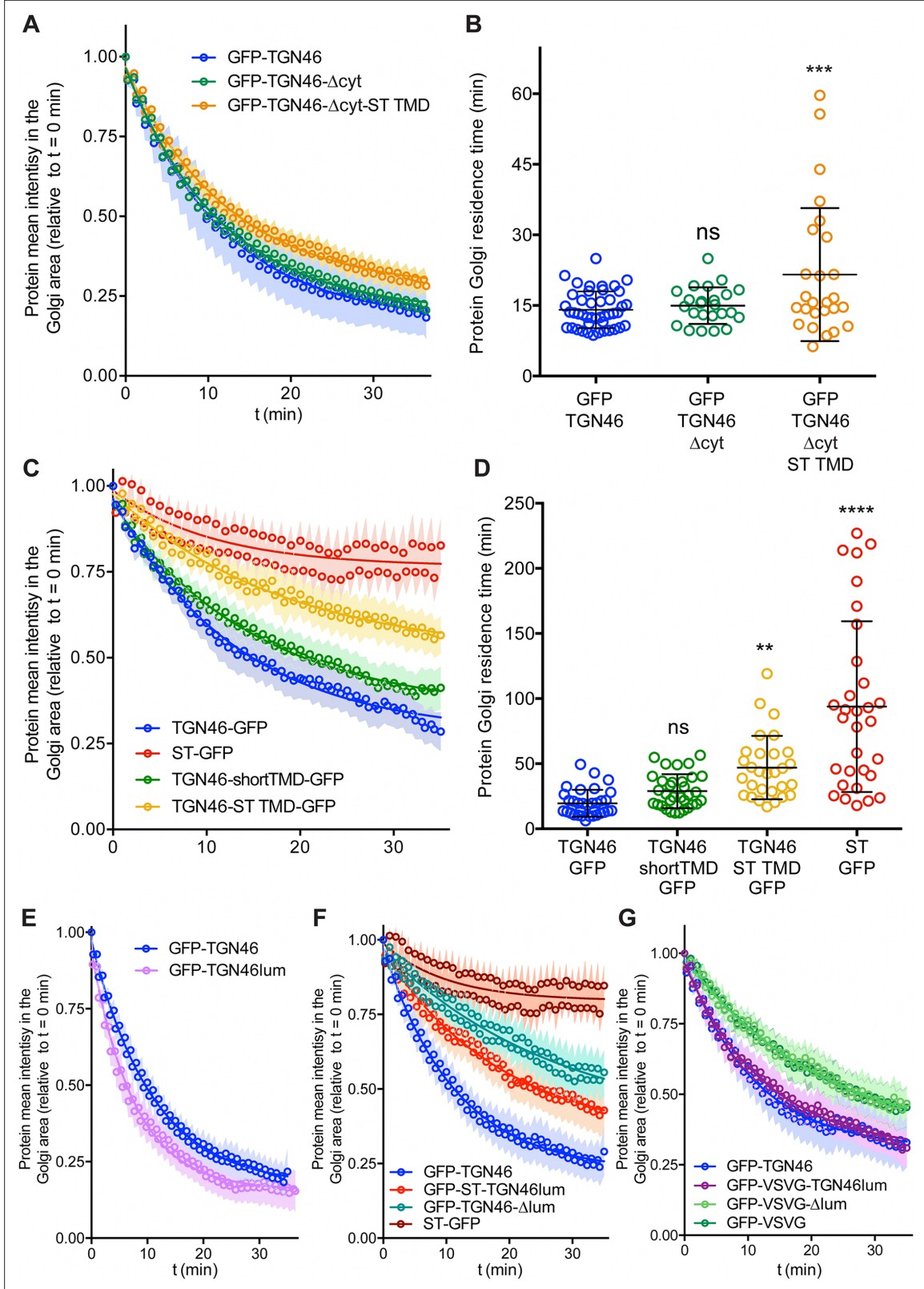

**Figure 3.** Fluorescence loss in photobleaching (FLIP) experiments monitor Golgi residence times of different proteins. (**A, C, E–G**) Relative fluorescence intensity average time trace (mean ± standard error of the mean [s.e.m.]) of FLIP experiments for the indicated proteins. Symbols correspond to actual measurements, solid lines to the fitted exponential decays. (**B, D**) Residence time in the perinuclear area measured as the half time of the FLIP curves.

*Figure 3 continued on next page*

*Figure 3 continued*

Results are from 7 to 12 cells from each of $n$ = 3 independent experiments (individual values shown, with mean ± stdev; ns, p > 0.05; **p ≤ 0.01; ***p ≤ 0.001; ****p ≤ 0.0001).

The online version of this article includes the following figure supplement(s) for figure 3:

**Figure supplement 1.** Fluorescence loss in photobleaching (FLIP) experiments monitor Golgi residence times of different proteins.

in cells treated with short-chain ceramide (D-ceramide-C6), a treatment that causes morphological changes and the physical segregation of the Golgi membranes into distinct ST- and TGN46-positive Golgi subdomains (*Duran et al., 2012*; *van Galen et al., 2014*; *Campelo et al., 2017*; *Capasso et al., 2017*), we did not observe any apparent change in the localization of the TGN46 mutants with short TMD with respect to TGN46 WT (*Figure 4—figure supplement 1*). However, because our results were obtained using diffraction-limited confocal microscopy (with a spatial lateral resolution of ~250 nm), it is still possible that more subtle alterations in the intra-Golgi localization of these mutants exist at shorter length scales. To address this possibility, we used super-resolution single-molecule localization microscopy with a localization precision (resolution) of ~15–30 nm. Our results show that, even with this enhanced spatial resolution, the intra-Golgi localization of the different TGN46 mutants with shorter TMD was indistinguishable from that of TGN46 WT (*Figure 4—figure supplement 2*).

Second, we asked whether the TGN46 TMD sequence plays a role in dragging this protein into CARTS. Interestingly, the two mutants with shorter TMDs (TGN46-shortTMD and TGN46-ST TMD) lost their preference for being co-packaged with PAUF into CARTS, and partially redirected into other transport carriers such as VSVG carriers (*Figure 4D, E*), even though they keep a similar steady-state intra-Golgi localization as the TGN46 WT (*Figure 4A*).

Finally, we investigated how fast these TGN46 mutants with shorter TMDs are exported out of the Golgi apparatus. FLIP experiments indicated that the chimeric protein TGN46-ST TMD-GFP has a slower Golgi export rate (longer Golgi residence time) as compared to TGN46-GFP. In addition, TGN46-shortTMD-GFP was exported at a slightly slower rate, although this difference was not statistically significant (*Figure 3C, D* and *Figure 3—figure supplement 1B*). Next, we co-overexpressed TGN46-GFP together with TGN46-ST TMD-mRFP or TGN46-mRFP, as a control, and performed FLIP experiments to measure the rate of TGN46 WT-GFP export. The aim of these experiments was to test if the expression of these mutants leads to defects in CARTS biogenesis. Our results show that the Golgi residence time of TGN46-GFP was not affected by the co-expression of either the WT or the chimeric proteins (*Figure 4—figure supplement 3A, B*). These data provide further support to the idea that the slower Golgi export rate of TGN46 mutants with short TMDs is a consequence of their compromised selective sorting into CARTS without detectable changes in their intra-Golgi localization.

It has been reported that some medial Golgi enzymes can dimerize or oligomerize and that this property (kin recognition) is important for TM protein localization along the Golgi stack (*Welch and Munro, 2019*; *Banfield, 2011*; *Lujan and Campelo, 2021*). However, there are reports showing that kin recognition is not relevant for ST or for other *trans*-Golgi membrane proteins (*Munro, 1995*). To examine if the unaltered intra-Golgi localization of TGN46-shortTMD and TGN46-ST TMD compared to WT was due to protein dimerization/oligomerization, we tested whether TGN46-GFP and TGN46-mRFP co-immunoprecipitated when expressed in HeLa cells. Our results show no obvious dimerization between TGN46-GFP and TGN46-mRFP (*Figure 4—figure supplement 3C*).

One of the possible consequences of shortening the TMD of TGN46 might be the partial burial of the highly charged residues of TGN46 cytosolic tail into the cytosolic leaflet of the TGN membrane, which, concomitantly, could negatively hamper its CARTS-mediated export. If so, the observed phenotype would follow from the unnatural proximity of the cytosolic tail to the membrane rather than from a direct effect due to hydrophobic mismatch. To test this hypothesis, we deleted the cytosolic tail of TGN46-ST TMD – which, as we showed earlier (*Figure 2*), has no major effect in TGN46 export in CARTS – to generate a mutant with TGN46 lumenal domain, ST TMD, and no cytosolic tail (TGN46-Δcyt-ST TMD, *Figure 2A*). This chimeric protein phenocopies GFP-TGN46-Δcyt regarding its (1) intra-Golgi localization (*Figure 2B, C* and *Figure 4B, C*), and (2) CARTS specificity (*Figure 2D, E* and *Figure 4D, E*), with only a milder reduction in the Golgi export rate (*Figure 3A, B* and *Figure 3—figure supplement 1A*). Taken together, these results strongly indicate that the hydrophobic matching

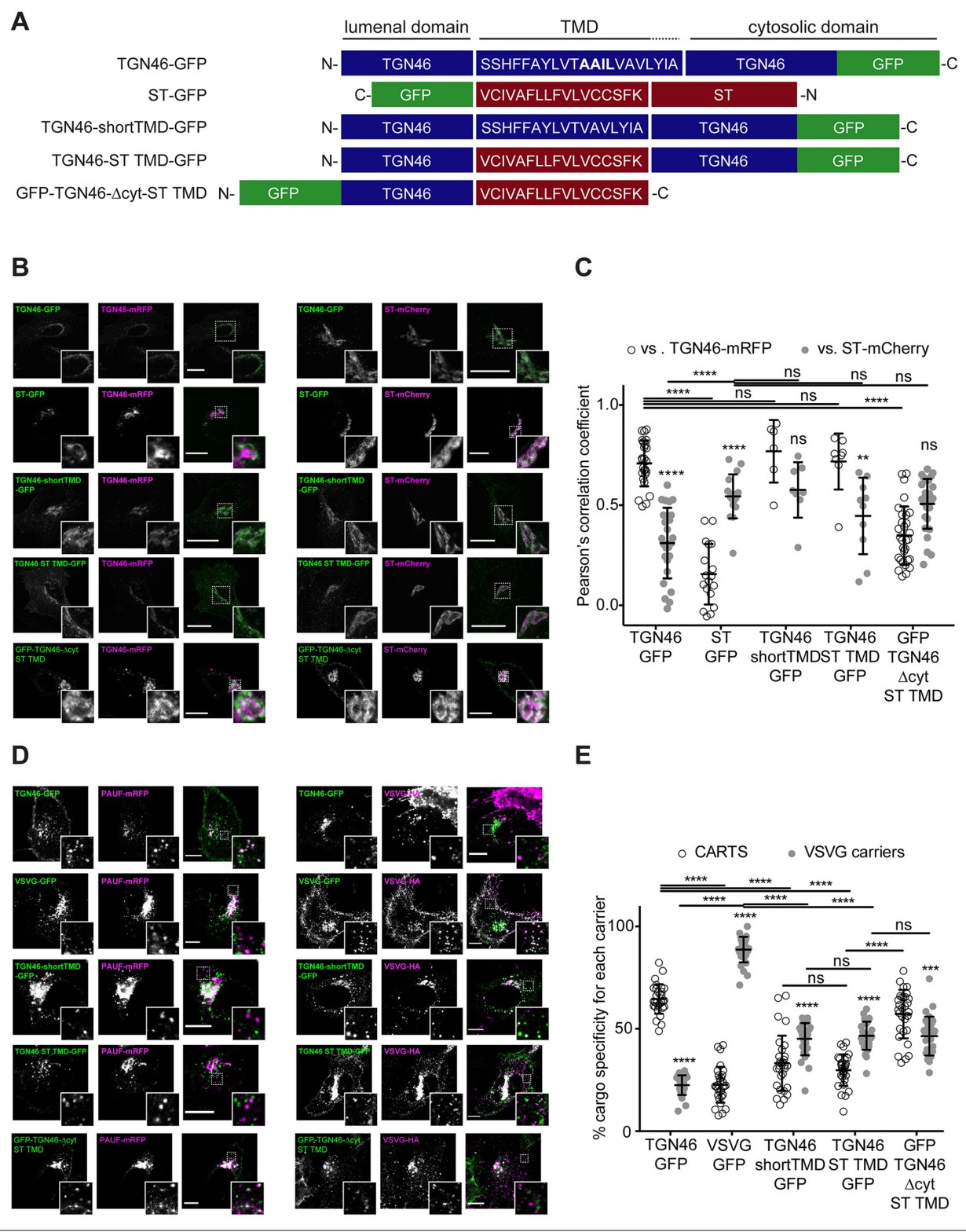

**Figure 4.** TGN46 intra-Golgi localization and CARTS specificity are insensitive to transmembrane domain (TMD) length and composition. (**A**) Schematic representation of construct domain topology. The amino acid sequence (in the correct topology) of the different TMDs is indicated. (**B**) HeLa cells co-expressing the different indicated proteins (green and magenta channels) were fixed, and the localization of those proteins was monitored by fluorescence confocal microscopy. Insets correspond to zoom-in areas of the dashed, white boxed areas. (**C**) Pearson's correlation coefficient between the perinuclear fluorescence signal of the *x*-axis indicated proteins with respect to TGN46-mRFP (empty circles) or ST-mCherry (gray circles), measured

*Figure 4 continued on next page*

*Figure 4 continued*

from confocal micrographs in (**B**). Results are at least three cells from each of *n* = 3 independent experiments (individual values shown, with mean ± stdev; ns, p > 0.05; **p ≤ 0.01; ****p ≤ 0.0001). (**D**) HeLa cells co-expressing the different indicated proteins (green and magenta channels) were fixed, processed for immunostaining when required, and the localization of those proteins was monitored by fluorescence confocal microscopy. Insets correspond to zoom-in areas of the dashed, white boxed areas. (**E**) Percentage of transport carriers containing each of the cargoes described on the *x*-axis that are also positive for pancreatic adenocarcinoma upregulated factor (PAUF; CARTS, empty circles) or VSVG (VSVG carriers, gray circles), as measured from confocal micrographs in (**D**). Results are from at least 10 cells from each of *n* = 3 independent experiments (individual values shown, with mean ± stdev; ns, p > 0.05; ***p ≤ 0.001; ****p ≤ 0.0001). Scale bars in (**B, D**) are 10 μm.

The online version of this article includes the following source data and figure supplement(s) for figure 4:

**Figure supplement 1.** Effect of sphingolipid metabolism on the intra-Golgi localization of TGN46 mutants.

**Figure supplement 2.** STORM analysis of intra-Golgi localization of different TGN46 mutants.

**Figure supplement 3.** Co-expression of different TGN46 proteins does not affect CARTS biogenesis or cargo export rate.

**Figure supplement 3—source data 1.** Uncropped images of the membranes of the Western blotting shown in *Figure 4—figure supplement 3C*.

mechanism (as well as the information encoded in the TMD) does not play a primary role in driving the intra-Golgi localization and Golgi export of TGN46.

## The lumenal domain of TGN46 is necessary and sufficient for CARTS-mediated export from the TGN

Finally, we enquired if the lumenal domain of TGN46 plays a role in intra-Golgi localization, Golgi export rate (residence time), and selective sorting into CARTS. First, we aimed to test whether the lumenal domain of TGN46 is *necessary* for its sorting and packaging into CARTS. We thus generated a deletion mutant of TGN46 lacking its lumenal domain (GFP-TGN46-Δlum) (*Figure 5A*), expressed it in parental HeLa cells, and monitored its intra-Golgi localization by fluorescence microscopy. Although still perinuclear, GFP-TGN46-Δlum lost the characteristic intra-Golgi localization of the TGN46 WT and rather localized into ST-mCherry-positive processing domains of the *trans*-Golgi membranes (*Figure 5B, C*). Notably, this exclusion from TGN export domains paralleled a striking reduction in the selective incorporation of GFP-TGN46-Δlum into CARTS (*Figure 6A, B*), as well as in the rate of Golgi export (*Figure 3F* and *Figure 6—figure supplement 1A, C*), as compared to the full-length protein. These results consistently indicate that the lumenal domain of TGN46 is *necessary* for CARTS-mediated export of TGN46, therefore suggesting that this domain contains a signal for sorting and loading into CARTS.

Next, we asked whether the TGN46 lumenal domain is not only necessary but also *sufficient* for CARTS-mediated TGN exit. We started by generating a soluble cargo protein that only contains the lumenal domain of TGN46, tagged with GFP at the N terminus just following the signal sequence (GFP-TGN46lum) (*Figure 5A*). Upon expression in HeLa cells, GFP-TGN46lum localizes to the Golgi apparatus where it only shows a partial colocalization with TGN46-mRFP or with ST-mCherry (*Figure 5B, C*). This low degree of colocalization could be due to the fact that we are comparing the location of soluble and TM proteins. Nonetheless, when we used our synchronized cargo release assay to measure selective incorporation into CARTS, GFP-TGN46lum maintained specificity for CARTS-mediated export (*Figure 6A, B*). Moreover, FLIP experiments revealed that GFP-TGN46lum exits as fast as GFP-TGN46 WT, or even faster (*Figure 3E* and *Figure 6—figure supplement 1A, B*). We next asked if the signal encoded in the TGN46 lumenal domain can direct otherwise unrelated proteins into CARTS. In particular, we tested whether TGN46 lumenal domain can lead to CARTS-mediated export of (1) a Golgi-resident TM protein (ST), and (2) a CARTS-independent TM cargo protein (VSVG). To that end, we exchanged the lumenal domains (while maintaining the correct topology) of ST and VSVG for that of TGN46, thereby generating two chimeric constructs: GFP-ST-TGN46lum, and GFP-VSVG-TGN46lum, respectively (*Figure 5A*). We then studied the intra-Golgi localization, CARTS-mediated export selectivity, and Golgi export rate of these proteins in relation to their WT counterparts. Our results indicate that GFP-ST-TGN46lum maintained a good degree of colocalization with ST-mCherry and additionally it increased its colocalization with TGN46-mRFP (*Figure 5B, C*). Similar results were obtained for the intra-Golgi localization of GFP-VSVG-TGN46lum with respect to TGN46-mRFP and VSVG-HA (*Figure 5D, E*). As an additional control to eliminate any sources of possible intra-Golgi sorting signals residing in the lumenal region of VSVG, we also included GFP-VSVG-Δlum – a deletion

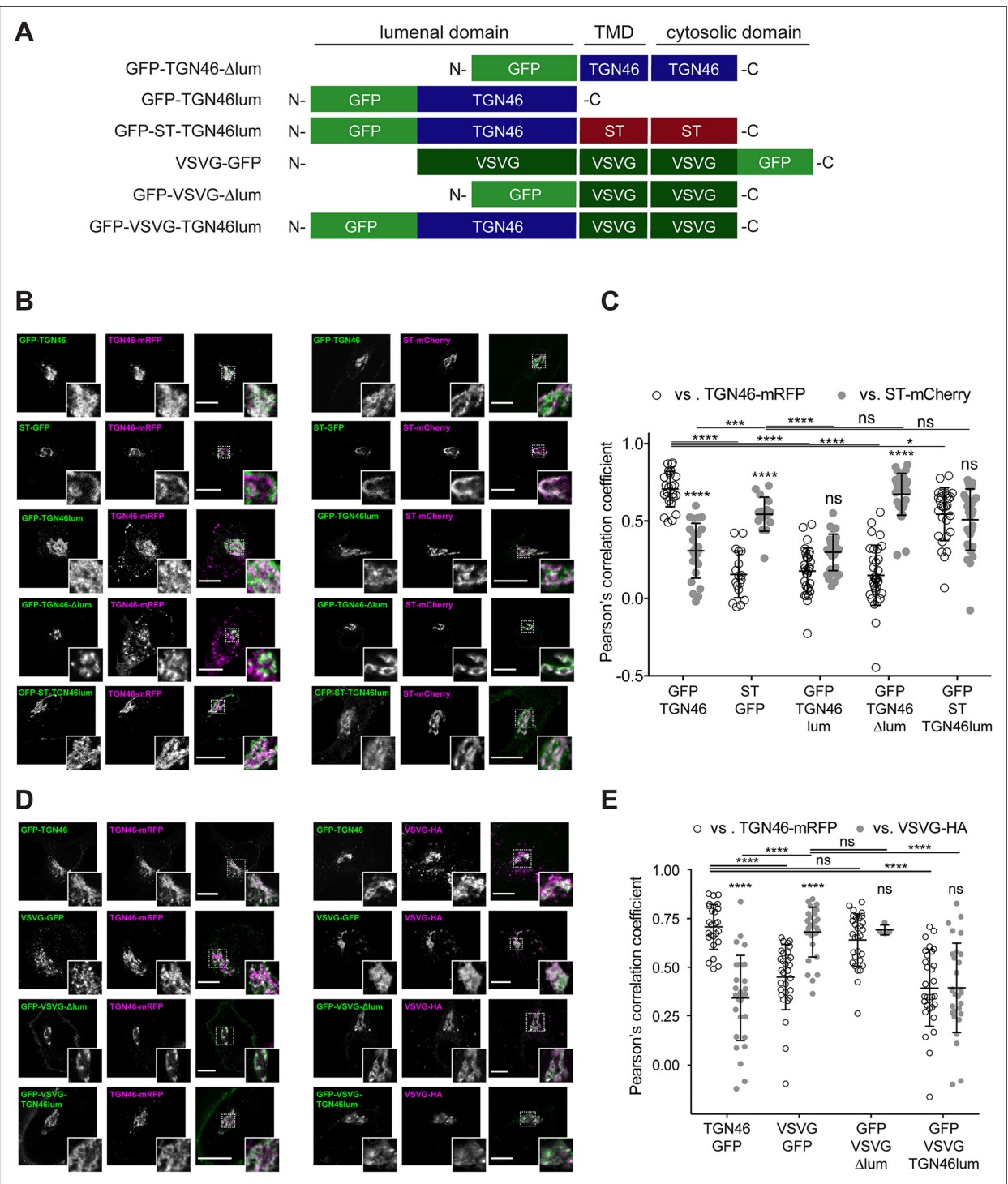

**Figure 5.** Intra-Golgi localization analysis for the role of the lumenal domain of TGN46 in CARTS-mediated export from the *trans*-Golgi network (TGN). (**A**) Schematic representation of construct domain topology. TMD: transmembrane domain. (**B**) HeLa cells co-expressing the different indicated proteins (green and magenta channels) were fixed, and the localization of those proteins was monitored by fluorescence confocal microscopy. Insets correspond to zoom-in areas of the dashed, white boxed areas. (**C**) Pearson's correlation coefficient between the perinuclear fluorescence signal of the *x*-axis indicated proteins with respect to TGN46-mRFP (empty circles) or ST-mCherry (gray circles), measured from confocal micrographs in (**B**). Results are from at least 10 cells from each of *n* = 3 independent experiments (individual values shown, with mean ± stdev; ns, p > 0.05; *p ≤ 0.05; ***p ≤ 0.001; ****p ≤ 0.0001). (**D**) HeLa cells co-expressing the different indicated proteins (green and magenta channels) were fixed, processed for immunostaining when required, and the localization of those proteins was monitored by fluorescence confocal microscopy. Insets correspond to zoom-in areas of the dashed, white boxed areas. (**E**) Pearson's correlation coefficient between the perinuclear fluorescence signal of the *x*-axis-indicated proteins with respect

*Figure 5 continued*

to TGN46-mRFP (empty circles) or VSVG-HA (gray circles; detected by immunofluorescence using an Alexa Fluor 647-conjugated secondary antibody), measured from confocal micrographs in (**D**). Results are from at least 10 cells from each of *n* = 3 independent experiments (individual values shown, with mean ± stdev; ns, p > 0.05; ****p ≤ 0.0001). Scale bars in (**B, D**) are 10 µm.

mutant of VSVG lacking the lumenal domain (*Figure 5A*). This construct shows a reduced colocalization with respect to both TGN46-mRFP and VSVG-HA, possibly indicating a tug-of-war between lumenal signals – driving localization toward TGN46-positive export domains – and TM/cytosolic signals – driving localization toward VSVG-positive export domains (*Figure 5D, E*). Despite these subtle differences in their intra-Golgi localization, the GFP-VSVG-TGN46lum chimera specifically exits the TGN in CARTS, with a similar specificity to that of TGN46-GFP, whereas GFP-VSVG-Δlum exits in VSVG carriers, similarly to VSVG-GFP (*Figure 6A, B*). In line with these observations, FLIP microscopy showed a fast Golgi export rate of GFP-VSVG-TGN46lum, indistinguishable from that of GFP-TGN46 WT, and a somewhat slower export rate for both GFP-VSVG-Δlum and VSVG-GFP (*Figure 3F, G* and *Figure 6—figure supplement 1A, C, D*). These results indicate that the lumenal domain of TGN46 can re-direct other TM cargo proteins into CARTS, thereby acting as a gain-of-function domain for fast export in CARTS out of the Golgi apparatus. Similarly, fusing the lumenal domain of TGN46 to ST also created an export-competent protein that exits the TGN by being specifically sorted into CARTS (*Figure 6A, B*). Remarkably, GFP-ST-TGN46lum has a shorter Golgi residence time as compared to ST-GFP, with an export rate slightly slower but comparable to that of GFP-TGN46 WT (*Figure 3F* and *Figure 6—figure supplement 1A, C*). Altogether, these results indicate that, as long as the protein is correctly localized within the Golgi membranes, signals encoded in the lumenal domain of TGN46 are not only necessary, but also *sufficient* for cargo sorting into CARTS.

## Correlation between CARTS specificity and Golgi export rate of cargo proteins

Our data thus far suggest that proteins with a high specificity for the CARTS export pathway have a relatively fast Golgi export rate (*Figures 2–6*). To quantitatively illustrate these findings, we plotted for each of the tested cargo proteins their CARTS specificity as a function of their Golgi residence time. The plots reveal the existence of a negative correlation between CARTS specificity and Golgi residence time (slope ~−0.9 ± 0.4% min⁻¹, $r^2$ ~ 0.4) (*Figure 7A*). Both TGN46 WT (*Figure 7A*, dark green) and the different chimeric cargo proteins that contain the lumenal domain but not the cytosolic tail of TGN46 (*Figure 7A*, light green and light blue) are selectively sorted into CARTS (>50% specificity) and rapidly exported out of the Golgi (short Golgi residence times ~10–30 min). Interestingly, in the absence of the lumenal domain of TGN46 (*Figure 7A*, red) or when only the TMD sequence of TGN46 is altered (while keeping the WT cytosolic tail) (*Figure 7A*, yellow), cargoes lost CARTS specificity (<35%) with the concomitant decrease in the Golgi export rate (larger Golgi residence times ~30–50 min), consistent with the values found for VSVG (*Figure 7A*, blue), a cargo that is excluded from CARTS (*Wakana et al., 2012*) and shows a slower Golgi export rate (Golgi residence time ~30 min). Taken together, these data indicate that VSVG-containing carriers mediate a relatively slower Golgi export route as compared to CARTS, thereby underscoring the different export kinetics of distinct Golgi export routes.

## The cargo sorting function of TGN46 is mediated by its lumenal domain

TGN46 is a bona fide CARTS component (*Wakana and Campelo, 2021*), and our data indicate that it is required for PAUF sorting into CARTS for secretion (*Figure 1*). Furthermore, we have shown that the lumenal domain of TGN46 is necessary and sufficient for the incorporation of this TM protein into CARTS (*Figures 5 and 6*). Thus, it is plausible that TGN46 facilitates the sorting of client cargoes via its lumenal domain. To test this hypothesis, we investigated whether the expression of different mutants of TGN46 rescues the sorting and export defect we have observed in TGN46-KO cells. In particular, we studied the following TGN46-based proteins: GFP-TGN46 (wild type protein), GFP-TGN46-Δcyt (no cytosolic tail), GFP-TGN46-ST TMD (TGN46 with the shorter TMD of the Golgi-resident ST), and GFP-TGN46-Δlum (no lumenal domain). First, TGN46-KO cells co-expressing any of these TGN46 constructs together with PAUF-mRFP or VSVG-HA were synchronized by a 20°C block and 15 min release at 37°C, after which the cells were fixed and prepared for fluorescence microscopy (*Figure 7—figure*

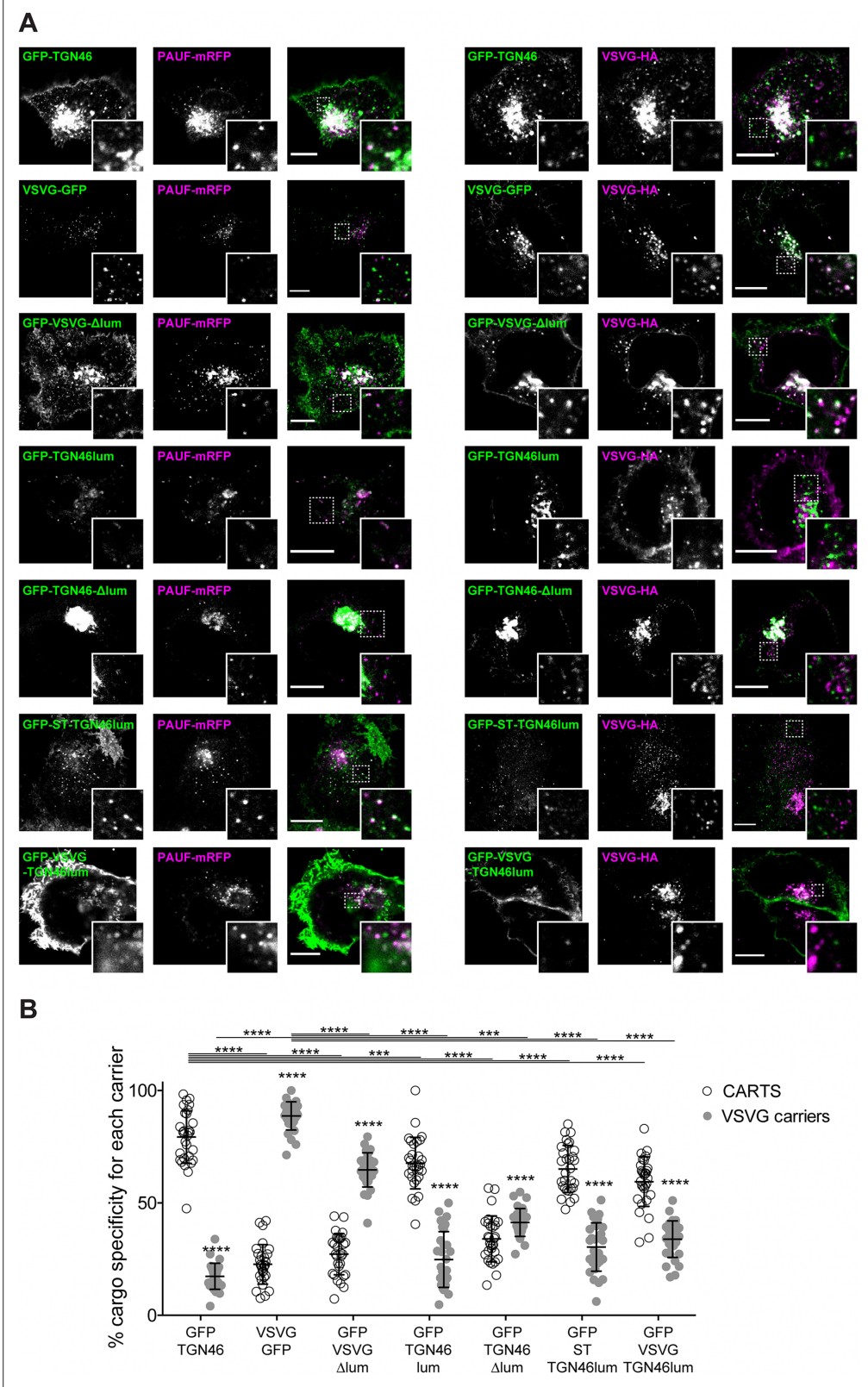

**Figure 6.** The lumenal domain of TGN46 is necessary and sufficient for its CARTS-mediated export from the *trans*-Golgi network (TGN). (**A**) HeLa cells co-expressing the different indicated proteins (green and magenta channels) were fixed, processed for immunostaining when required, and the localization of those proteins was monitored by fluorescence confocal microscopy. Insets correspond to zoom-in areas of the dashed, white boxed areas. Scale

*Figure 6 continued on next page*

*Figure 6 continued*

bars are 10 µm. (**B**) Percentage of transport carriers containing each of the cargoes described on the *x*-axis that are also positive for pancreatic adenocarcinoma upregulated factor (PAUF; CARTS, empty circles) or VSVG (VSVG carriers, gray circles), as measured from confocal micrographs in (**A**). Results are from at least 10 cells from each of *n* = 3 independent experiments (individual values shown, with mean ± stdev; ***p ≤ 0.001; ****p ≤ 0.0001).

The online version of this article includes the following figure supplement(s) for figure 6:

**Figure supplement 1.** The lumenal domain of TGN46 is necessary and sufficient for its CARTS-mediated export from the *trans*-Golgi network (TGN).

*supplement 1A*). We quantified the percentage of carriers containing the different TGN46 mutants that were also positive for either PAUF-mRFP (CARTS marker) or VSVG-HA (non-CARTS marker) and compared those to the distributions found when the same proteins were expressed in the parental HeLa cells (*Figure 7B*; see also *Figures 2E, 4E, and 6B*). Our results indicate that CARTS selectivity of these cargo proteins was not severely altered in the TGN46 KO cells as compared to the parental cell line. Next, we carried rescue experiments by performing FLIP in TGN46-KO cells expressing PAUF-mRFP together with GFP-TGN46, GFP-TGN46-Δcyt, TGN46-ST TMD-GFP, or GFP-TGN46-Δlum to measure the rate of PAUF-mRFP export from the Golgi. Interestingly, overexpression of GFP-TGN46 WT or GFP-TGN46-Δcyt – but not of TGN46-ST TMD-GFP or GFP-TGN46-Δlum – was able to rescue PAUF-mRFP export from the Golgi in TGN46-KO cells (*Figure 7C, D* and *Figure 7—figure supplement 2*). Taken our data together, we propose a model in which the proper partitioning of TGN46 into TGN export domains (sites of CARTS biogenesis) is required for its lumenal domain to efficiently recruit and sort PAUF and possibly other CARTS cargoes into nascent transport carriers for secretion.

## Discussion
### TGN46 is necessary for cargo loading into CARTS

Here, we revealed a novel role for TGN46 in the sorting of the secretory cargo protein PAUF into nascent transport carriers at the TGN. CRISPR/Cas9-edited HeLa cells lacking TGN46 showed (1) a drastic inhibition in PAUF secretion (*Figure 1A*); (2) a reduction in the number of cytoplasmic CARTS – but not of other TGN-to-PM carriers such as VSVG carriers – (*Figure 1B*); and (3) a decrease in the Golgi export rate of PAUF (*Figure 1C, D*). To investigate the role of TGN46 in TGN export, we took advantage of the phenotype created by the exogenous expression of PKD-KD, which consists in the presence of long, cargo-containing tubules emanating from the TGN membranes (*Liljedahl et al., 2001*). Because these tubules are fission-defective transport carriers, and membrane fission is downstream of cargo sorting, we could use this assay as a useful approach to investigate cargo sorting into PKD-dependent transport carriers at the TGN. Our results showed that TGN46 is not necessary for membrane tubulation but it is essential to sort and package the secretory cargo PAUF into PKD-KD-induced TGN tubules (*Figure 1E*), therefore suggesting a prime role of TGN46 in cargo sorting. Indeed, TGN46 (and its rodent orthologs TGN38/TGN41) had previously been suggested to act as a cargo receptor, based on the fact that this protein shares the characteristics of a typical cargo receptor: it is a type-I single-pass TM protein that cycles between the TGN and the PM (*Stanley and Howell, 1993*). Although future work will be necessary to pinpoint the mechanism that TGN46 uses for cargo sorting and loading into carriers, we believe that the data reported here shed some light into the actual role of human TGN46 for the sorting and loading of cargo proteins into TGN-derived transport carriers destined to the cell surface.

It is important to underscore that our findings using the TGN46 KO cell line reveal a partial inhibition rather than a complete block in PAUF export (*Figure 1A, C, D* and *Figure 7C, D*). This incomplete block may stem from various factors. *First*, cells may respond to genetic perturbations, such as those induced by a gene knockout, by adapting to compensate the loss of a specific gene. These compensatory mechanisms could potentially mitigate the full impact of TGN46 depletion, providing a plausible explanation for the observed partial effects. *Second*, our data indicate that the absence of TGN46 reduces PAUF secretion without completely inhibiting its export (*Figure 1A*). These results align with our proposed role for TGN46 in cargo sorting. In the absence of TGN46, secretory cargo proteins might find alternative, TGN46-independent routes for export, such as through bulk flow or uncontrolled incorporation into other transport carriers. Indeed, the partial redistribution of the

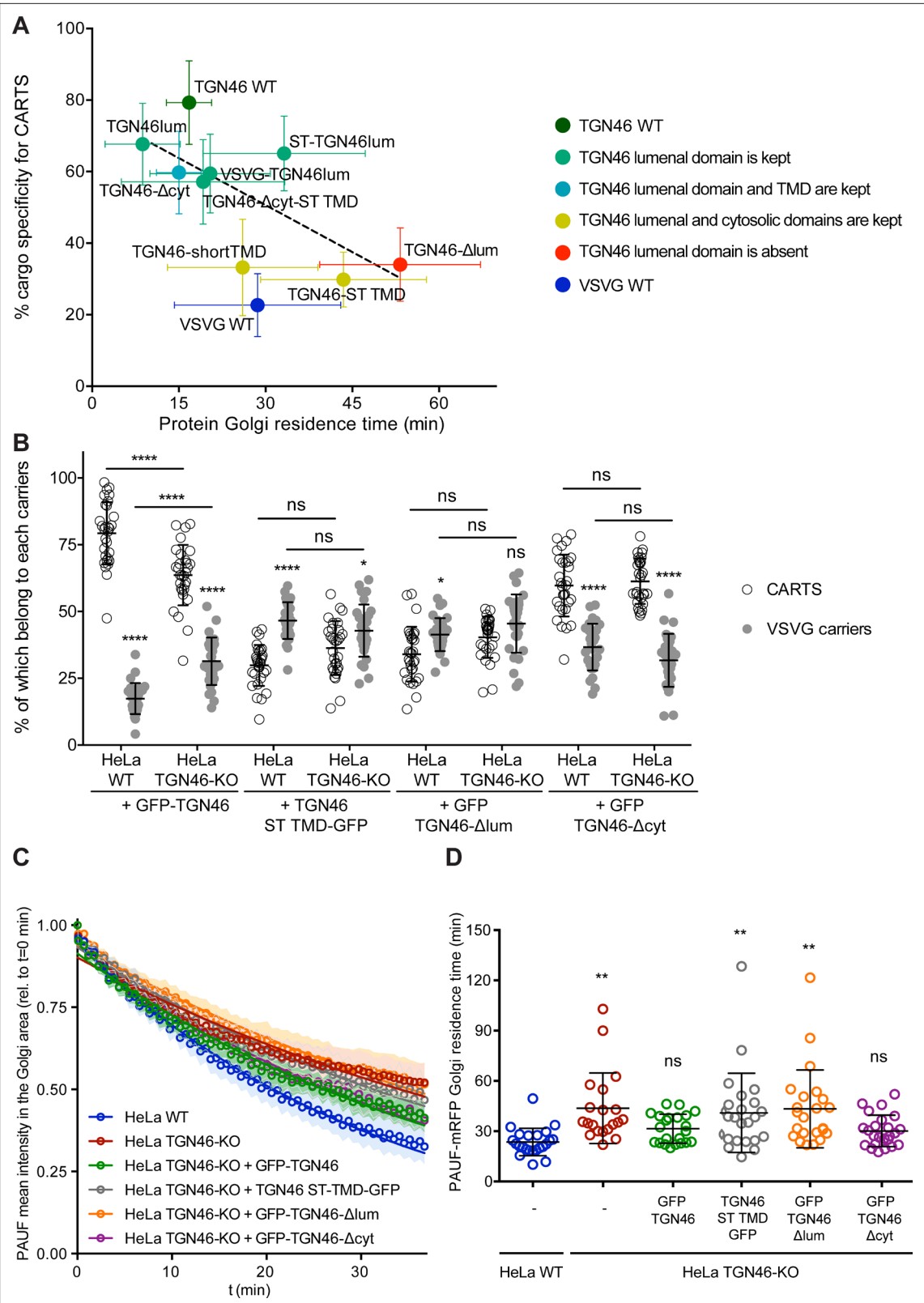

**Figure 7.** The cargo sorting function of TGN46 is mediated by its lumenal domain. (**A**) CARTS specificity of cargo proteins correlates with their Golgi export rate. Plot of the percentage of cargo-positive vesicles that are also positive for pancreatic adenocarcinoma upregulated factor (PAUF; CARTS marker) as a function of the Golgi residence time as measured from fluorescence loss in photobleaching (FLIP) experiments, for the different indicated cargo proteins (color coding explained in the legend on the right). Dashed black line represents a linear fit of the data points (shown as mean ±

*Figure 7 continued on next page*

*Figure 7 continued*

standard error of the mean [s.e.m.]), where the slope is statistically different from zero (extra sum-of-squares *F* test, p-value = 0.04). (**B**) Percentage of transport carriers containing each of the cargoes described on the *x*-axis that are also positive for PAUF (CARTS, empty circles) or VSVG (VSVG carriers, gray circles), as measured from confocal micrographs of HeLa cells (either WT or TGN46-KO cell lines) expressing the indicated proteins. Results are at least 10 cells from each of *n* = 3 independent experiment (individual values shown, with mean ± stdev; ns, p > 0.05; *p ≤ 0.05; ****p ≤ 0.0001). (**C**) Relative fluorescence intensity average time trace (mean ± s.e.m.) of FLIP experiments for the indicated proteins expressed in HeLa WT or HeLa TGN46-KO cells, as detailed in the legend. Symbols correspond to actual measurements, solid lines to the fitted exponential decays. (**D**) Residence time of PAUF-mRFP in the perinuclear area of HeLa cells (WT or KO), expressing the different proteins as labeled in the *x*-axis, and measured as the half time of the FLIP curves. Results are from 7 to 12 cells from each of *n* = 3 independent experiments (individual values shown, with mean ± stdev; ns, p > 0.05; **p ≤ 0.01).

The online version of this article includes the following figure supplement(s) for figure 7:

**Figure supplement 1.** The cargo sorting function of TGN46 is mediated by its lumenal domain.

**Figure supplement 2.** Golgi export of PAUF-mRFP is dependent on the lumenal domain of TGN46.

**Figure supplement 3.** TGN46 domain structure and sequence characteristics.

TGN46-Δlum mutant into VSVG carriers (*Figure 6B*) supports this possibility. Notably, similar situations have been observed when unrelated sorting factors were depleted from the Golgi membranes. For instance, disruption of the cofilin/SPCA1/Cab45 sorting pathway inhibits the secretion of pathway clients but does not bring this effect to a complete halt (*von Blume et al., 2011*; *von Blume et al., 2012*). And *third*, it is possible that TGN46 is not the sole player in the sorting of PAUF into CARTS. The presence of redundant or alternative mechanisms cannot be ruled out.

Although in here we have focused on the study of the model CARTS cargo protein PAUF, it seems reasonable to expect that TGN46 is responsible for the sorting of other CARTS cargoes. Interestingly, the lumenal domain of TGN38 has been shown to interact with the adhesion receptor integrin $\beta_1$ and to partially co-localize with integrins $\alpha_5$ and $\beta_1$, but not $\alpha_V$ or $\beta_3$ (*Wang and Howell, 2000*). In addition, TGN38 regulates cell adhesion, as the overexpression of TGN38 leads to cell detachment from the substrate (*Luzio et al., 1990*; *McNamara et al., 2004*; *Lee and Banting, 2002*; *Humphrey et al., 1993*; *Reaves et al., 1993*). Remarkably, integrin $\beta_1$ is transported from the TGN to the cell surface in a Rab6-mediated (*Huet-Calderwood et al., 2017*) and PKD-dependent manner (*Yeaman et al., 2004*). Because CARTS are PKD-dependent transport carriers that share many similarities with Rab6-positive carriers (*Wakana and Campelo, 2021*), we propose that the role of TGN46 as a cargo receptor for the sorting of the soluble CARTS-specific cargo PAUF is quite general, and can include a long list of both soluble and TM cargo proteins, such as $\alpha_5\beta_1$ integrins. Interestingly, TGN46 binds a cytosolic protein complex made of 62 kDa protein (p62) and Rab6 (*Jones et al., 1993*). These authors used a Golgi budding assay to test the involvement of the cytosolic domains of TGN38 and TGN41 in budding of Golgi-derived carriers that contain the TM cargo protein pIgA-R (polymeric IgA-receptor). The authors showed that the budding of these carriers was blocked upon incubation of the Golgi membranes with peptides against the cytosolic tail of TGN38/41. These data complements and contrasts our data showing that the cytosolic tail of TGN46 is dispensable for PAUF sorting into CARTS. It thus appears possible that TGN46 plays a multifaceted role in TGN export by controlling the export of soluble and TM cargo proteins through alternative mechanisms.

In addition, TGN46 has been recently reported to assist in the trafficking of severe acute respiratory syndrome coronavirus 2 (SARS-CoV-2), as this protein is upregulated in SARS-CoV-2-infected cells and cells depleted of TGN46 showed a reduced SARS-CoV-2 infection rate (*Zhang et al., 2022*). In summary, further work will be required to grasp the full magnitude of the role of TGN46 in protein sorting for export and secretion out of the TGN, which can provide ideas to help in the rational design of new therapeutic tools to target human pathologies associated with secretion defects.

## TGN46 localization to *trans*-Golgi membranes is independent of TMD length and composition

Different mechanisms that mediate the spatial and temporal organization of Golgi residents (e.g., glycosylation enzymes) and their respective substrates (e.g., secretory cargoes) have been proposed (*Banfield, 2011*; *Welch and Munro, 2019*; *Lujan and Campelo, 2021*). Among these, a lipid-based retention/sorting mechanism has been proposed for Golgi TM proteins (*Patterson et al., 2008*; *Munro, 1991*; *Sharpe et al., 2010*; *Munro, 1995*; *Lippincott-Schwartz and Phair, 2010*). This

mechanism builds on the observation that the levels of certain lipids, such as cholesterol and SM, are enriched along the *cis*- to *trans*-axis of the Golgi stack (*van Meer et al., 2008*; *Bretscher and Munro, 1993*). This lipid gradient correlates with an increase in the thickness of those membranes (*Mitra et al., 2004*). In addition, a comprehensive bioinformatic study using large datasets of TM proteins from different organisms showed a correlation between the intracellular localization of these proteins and different specific features of their TMDs, such as their length, which on average is shorter for proteins of the early secretory pathway as compared to those of the late secretory pathway (*Sharpe et al., 2010*). Furthermore, it has been proposed that not only the TMD length but also the distribution of bulky and highly hydrophobic amino acids within the TMD affect the specific location of TM proteins (*Sharpe et al., 2010*; *Quiroga et al., 2013*). Altogether, these studies have led to the suggestion of the hydrophobic matching model, which proposes that gradients in bilayer thickness along the membranes of the secretory pathway contribute/drive the sorting and retention of TM proteins across the secretory pathway. Although there is compelling evidence for a TMD-based mechanism for targeting certain TM proteins to specific cell surface domains (*Milovanovic et al., 2015*), to localize type-II resident proteins of the cell surface and the Golgi membranes (*Munro, 1995*; *Quiroga et al., 2013*), and also to sort the type-III TM protein linker for activation of T-cells (LAT) by raft-mediated partitioning (*Diaz-Rohrer et al., 2014*; *Lorent et al., 2017*; *Castello-Serrano et al., 2023*), it remained unclear how this mechanism contributed to the retention, sorting and lateral segregation of Golgi-resident enzymes and their substrates. Interestingly, by using a coarse-grained mathematical model of intra-Golgi transport, *Dmitrieff et al., 2013* showed that experimental data on the Golgi dynamics of secretory cargoes and Golgi residents cannot be explained if the hydrophobic matching mechanism is the sole retention mechanism for Golgi residents. In agreement with this latter proposal, our results revealed that hydrophobic mismatch between TMDs and bilayer thickness is not sufficient to drive the sorting of a TM Golgi-resident enzymes from substrate cargoes along the *trans*-Golgi/ TGN membranes. Specifically, we studied the role of TMD length and amino acid composition in intra-Golgi localization of the type-I single-pass TM cargo glycoprotein TGN46, and observed that localization and sialylation of TGN46 is not affected when its TMD is shortened or even replaced by that of the Golgi-resident type-II TM enzyme ST (*Figure 4A–C*). These data suggest that the hydrophobic length and amino acid composition of the TMD of TGN46 is not a major contributing factor to localize this TM protein into Golgi export domains for its function in sorting and packaging of soluble cargoes into nascent CARTS.

## A role for the cytosolic tail of TGN46 in driving its proper intra-Golgi localization

Our results indicate that the cytosolic tail of TGN46 contains a specific signal allowing the perinuclear pool of TGN46 to localize in the right sub-compartment within the TGN membranes. In particular, the localization of a TGN46 mutant lacking the cytosolic tail was shifted toward Golgi processing domains (ST enzyme-positive regions) rather than in export domains (TGN46 WT-positive regions) (*Figure 2B, C*). It has been well documented that the cytosolic tail of TGN38 – the rodent ortholog of TGN46 – contains a tyrosine-based motif for internalization and recycling of the protein back to the TGN (*Bos et al., 1993*; *Humphrey et al., 1993*). Adding to these earlier reports, our data suggest that the cytosolic tail of TGN46 also determines its fine intra-Golgi localization. Factors such as Vps74/ GOLPH3 or coat proteins such as COPI (*Tu et al., 2008*; *Wood et al., 2009*; *Welch et al., 2021*), influence enzyme–cargo interaction/segregation by recognizing cytoplasmic signals and thereby inducing protein retention by fast recycling. In addition, SM metabolism can also contribute to laterally organize the Golgi membranes at the nanoscale for efficient cargo–enzyme interactions (*Duran et al., 2012*; *van Galen et al., 2014*).

## Cargoes destined for secretion exit the Golgi with different export rates

We observed that the Golgi residence time of different TGN46-based chimeric proteins was different depending on the specific export route they use to exit the TGN (*Figure 7A*). In particular, when we compared the export rate of two different PKD-dependent TM proteins (TGN46, which is exported in CARTS; and VSVG, which is not), we found that CARTS appear to be a faster export route (~15 min Golgi residence time) than VSVG carriers (~30 min Golgi residence time) (*Figure 7A*). Remarkably,

when we generated a series of chimeras combining different domains from different proteins, we observed that loss of CARTS specificity correlated with larger Golgi residence times (*Figure 7A*). These results could provide an explanation as to why cells use different routes (with different machineries) for the export of secretory or TM cargo proteins to the same destination, that is, different cargoes might require longer/shorter residence times in the Golgi and/or faster transport kinetics to the cell surface. It is possible that developing various TGN-to-PM export routes of different kinetics might have endowed cells with beneficial mechanisms to cope with their secretory needs under stress conditions.

CARTS have been suggested to be enriched in cholesterol and SM (*Wakana et al., 2021*; *Wakana and Campelo, 2021*), and their biogenesis to require intact ER–Golgi membrane contact sites (MCS) supplying the TGN with these lipids (or their precursors) (*Wakana et al., 2015*; *Wakana et al., 2021*). Notably, data from Deng, Pakdel et al. suggested that TGN46 is a native cargo of the SM-dependent TGN export pathway, as it was identified by mass spectrometry to be co-packaged in carriers containing the lumenal SM sensor EQ-SM-APEX2 (*Deng et al., 2018*). Given the similarities between the SM/Cab45 and CARTS export pathways (*Stalder and Gershlick, 2020*; *Wakana and Campelo, 2021*), it is tempting to speculate that TGN46 could be enriched in ER–Golgi MCS and perhaps complement the sorting function of Cab45 for a subset of secretory cargo proteins.

Further elaborating on these specific needs/characteristics of different cargoes/export routes will be of outmost importance in the future to better understand the intricate relationships between physiological needs and the regulation of the secretory machinery of the cell (*Di Martino et al., 2019*; *Luini and Parashuraman, 2016*; *Cancino et al., 2013*; *Wakana and Campelo, 2021*).

## Ideas and speculations

In the following, we would like to propose some ideas and speculate on a possible mechanism by which TGN46 could serve as a cargo sorting factor at the TGN. Our data suggest that the lumenal domain of TGN46 might have a propensity to self-associate, because (1) the steady-state intra-Golgi localization of TGN46 is lost when the lumenal domain is deleted (*Figure 5B, C*); and (2) although the soluble lumenal domain of TGN46 showed an intermediate intra-Golgi localization phenotype, anchoring this domain to the membrane by fusing it to the *trans*-Golgi-targeting sequence of ST conferred this chimera a localization pattern indistinguishable from that of full-length TGN46 (*Figure 5B, C*). To obtain insights on how this might occur, we analyzed the sequence features of the lumenal domain of TGN46 (*Figure 7—figure supplement 3A*). On the one hand, TGN46 is a heavily glycosylated protein, including nine predicted *N*-linked glycosylation sites, multiple *O*-linked glycosylation sites, as well as five phosphosites (*Figure 7—figure supplement 3B*), all of them located at the lumenal domain. Structurally, this region is highly disordered, as evidenced by a high PONDR score (in the range 0.5–1) for most of its amino acid sequence (*Figure 7—figure supplement 3C*), indicating with high probability that the lumenal domain of TGN46 is an essentially intrinsically disordered region (IDR). In recent years, IDR of various proteins have been shown to have the capacity to form biomolecular condensates (*Banani et al., 2017*; *Garcia-Cabau and Salvatella, 2021*; *Shin and Brangwynne, 2017*; *Hyman et al., 2014*; *Brangwynne et al., 2009*; *Case et al., 2019*; *Mittag and Pappu, 2022*). Due to their liquid nature, these condensates take spherical shapes, and their components are mobile within the condensate (dense phase) and can also exchange with those present, at a lower concentration, in the external solution (light phase). Carefully testing whether the TGN46 lumenal domain has the capacity to form liquid droplets in vitro and in a cellular context will be necessary to hypothesize that condensate formation assisted by TGN46 lumenal domain could mediate the sorting of secretory cargoes into transport carriers for their secretion. Naturally, these ideas still remain highly speculative but they help us ask a number of interesting questions. First, what is the role of TGN46 lumenal domain glycosylation in its sorting function, and, potentially, in the hypothetical regulation of droplet/condensate formation? Interestingly, the levels of the cation-dependent mannose 6-phosphate receptor – a well-known cargo receptor at the TGN for the sorting of lysosomal hydrolases into clathrin-coated vesicles – are about 10-fold higher than those of TGN46: ~200,000 copies vs. ~30,000 copies, respectively, per HeLa cell (*Kulak et al., 2014*). Of notice, our co-immunoprecipitation experiments showed no evidence for strong TGN46–TGN46 protein interactions (*Figure 4—figure supplement 3C*), which agrees with the possibility of TGN46 forming biomolecular condensates that are stabilized by weak interactions. It is therefore tempting to speculate that the mechanism of cargo sorting by TGN46

might be different from that of a classical cargo receptor, such as the mannose 6-phosphate receptor, which binds its clients in an ~1:1 stoichiometry. It still remains unknown whether PAUF is recruited to nascent CARTS by the lumenal domain of TGN46. PAUF is a putative lectin, so it is possible that it interacts with TGN by recognizing the glycans on its lumenal domain (*Figure 7—figure supplement 3B*). Alternatively, or in parallel, TGN46 condensates could be liquid droplets with a specific chemical microenvironment (pH, $Ca^{2+}$ concentration, etc.), which might serve to recruit certain cargoes. Future endeavors along these lines will help elucidate the mechanistic details of cargo sorting by TGN46.

Remarkably, *Parchure et al., 2022* recently showed that liquid–liquid phase separation (LLPS) might play a dominant role in the formation of secretory insulin-containing granules for regulated exocytosis. The authors showed that the chromogranins – key regulators of secretory granule formation – can undergo LLPS in vitro in conditions of low pH, such as the one found in the TGN. Notably, cargoes such as proinsulin or LyzC do co-partition with chromogranins in liquid droplets, and put forward a 'client sponge' model in which secretory cargoes would be sorted into liquid biomolecular condensates based on their size.

## Conclusion

In summary, we presented here experimental results that support a model in which TGN46 serves as a sorting receptor for a subset of cargoes at the TGN. Thanks to TGN46, these cargoes are sorted packaged into nascent CARTS for their fast delivery to the cell surface or for secretion outside the cell. The sorting capabilities of TGN46 are ascribed to its lumenal domain, an IDR prone to undergo phase separation. Based on our findings, we propose a working model in which TGN46 functions to recruit secretory cargoes in the lumen of the TGN for their sorting and fast export to the cell surface by CARTS.

## Methods

### Reagents and antibodies

Brefeldin A (BFA), from Sigma-Aldrich, was dissolved in dimethyl sulfoxide (DMSO) (Sigma-Aldrich) as a 10 mg/ml stock solution, and used at 5 µg/ml concentration. *N*-Hexanoyl-D-erythro-sphingosine (D-cer-C6), from Matreya, was dissolved in pure ethanol (Merck) as a 10-mM stock solution, and used at 20 µM concentration. Cycloheximide was purchased from A.G. Scientific, diluted to 1 M in DMSO as a stock solution, and used at 100 µM concentration. Sheep anti-human TGN46 was from AbD Serotec. Mouse monoclonal against HA was from Biolegend. Mouse monoclonal against Myc was from Sigma-Aldrich. Rabbit polyclonal antibody against Flag was from Sigma-Aldrich. Rabbit polyclonal antibody against GFP was from Abcam. Alexa Fluor-labeled secondary antibodies were from Invitrogen and HRP-conjugated secondary antibodies from Sigma-Aldrich. For STORM, the activator/reporter dye-conjugated secondary antibodies were in-house-labeled donkey-anti-mouse and donkey-anti-rabbit obtained from ImmunoResearch, used at a final concentration of 20 µg/ml. The dyes used for labeling were NHS ester derivatives: Alexa Fluor 405 Carboxylic Acid Succinimidyl Ester (Invitrogen), Cy3 mono-Reactive Dye Pack (GE HealthCare), and Alexa Fluor 647 Carboxylic Acid succinimidyl Ester (Invitrogen). Antibody labeling reactions were performed as previously reported (*Borgman et al., 2020*).

### Cell culture, plasmids, and siRNA

HeLa cells (ATCC; negative mycoplasma tested) were cultured in Dulbecco's modified Eagle medium (DMEM; Lonza and Capricorn Scientific) containing 10% fetal bovine serum (FBS) (Thermo Fisher and Capricorn Scientific), 1% penicillin/streptomycin (Biowest), and 2 mM L-glutamine (Biowest). Cells were transfected with DNA using Xtreme-GENE 9 DNA (Roche) or with TransIT-HeLaMonster (Mirus) following the manufacturer's recommendations. The TGN46-GFP plasmid, which was kindly provided by Dr. S. Ponnambalam (Leeds University, Leeds, England, UK), was generated by inserting human TGN46 cDNA into a pEGFP-N1 vector using the BamHI restriction site. TGN46 ΔAAIL-GFP, TGN46-ST TMD-GFP, GFP-TGN46, GFP-TGN46 LT, GFP-TGN46 TC, GFP-TGN46lum, GFP-TGN46lum-ST TMD, GFP-TGN46lum-ST TC, GFP-TGN46lum-VSVG TC, and GFP-VSVG TC were sub-cloned by Gibson assembly (*Gibson et al., 2009*) and/or DNA digestion and ligation using TGN46-GFPDone as a backbone and TGN46-GFP, ST-GFP, or VSVG-GFP as DNA sequence providers. TGN46-mRFP was

generated by Gibson assembly of the KpnI- and EcoRI-digestion of the previously described plasmid PAUF-mRFP (*Wakana et al., 2012*). TGN46-ST TMD-mCherry was sub-cloned by Gibson assembly using TGN46-mRFP as the backbone and ST-GFP as DNA sequence providers. PAUF-MycHis and PAUF-mRFP plasmids were described earlier (*Wakana et al., 2012*). ST-GFP, which encodes for the initial 45 amino acids of the ST6Gal-I ST, comprising the cytosolic domain, the TMD and 20 amino acids of the lumenal domain, was cloned into pEGFP-N1 as previously described (*Duran et al., 2012*). The ST-mCherry plasmid was generated from the previously described plasmid ST-FKBP-mCherry (*Pecot and Malhotra, 2004*) by Gibson assembly of the BglII- and BamHI-digested plasmid. The VSVG-GFP plasmid was described previously (*von Blume et al., 2009*). VSVG-HA was generated by Gibson assembly from VSVG-GFP. PKD2-KD-Flag plasmid was described earlier (*Bossard et al., 2007*). pETM14-6his-mGFP-TGN46lum plasmid for bacterial expression was cloned using a synthetic fragment of bacterial optimized TGN46 (gblock IDT), meGFP amplified from GFP-TGN46-lum and assembled in pETM14 backbone vector using Gibson assembly. All plasmids were verified by sequencing (STAB vida). SnapGene software (obtained from GSL Biotech) and ApE software (by M. Wayne Davis) were used for molecular cloning procedures.

For TGN46 transient silencing, HeLa cells were transfected using HiPerFect Transfection Reagent (QIAGEN) with 5 nM of TGN46 siRNA (oligo name: SASI_HS01_00080765, from Sigma-Aldrich), following the manufacturer's recommendations. MISSION siRNA Universal Negative Control #1 was used as a control siRNA. 48 hr after transfection, corresponding experiments were performed.

Single-clone TGN46 KO HeLa cells were generated by CRISPR/Cas9. HeLa cells were transfected with U6-gDNA (5'-AAAGACGTCCCTAACAAGT-3'; clone ID es: HSPD0000063884): CMV-eCas9-2a-tGFP (Sigma-Aldrich). 48 hr after transfection, GFP-positive cells were sorted as individual cells by FACS (fluorescence-activated cell sorting ) using a BD Influx cell sorter (BD Biosciences). Upon clone expansion, KO effectiveness of the different clones was checked by sodium dodecyl sulfate–polyacrylamide gel electrophoresis (SDS–PAGE) and Western blotting and also by immunofluorescence microscopy.

Materials generated for this article will be available upon reasonable request.

## Immunofluorescence microscopy

Samples were fixed with 4% formaldehyde in phosphate-buffered saline (PBS) for 15 min and permeabilized and blocked with 0.2% Triton X-100, 3% bovine serum albumin (BSA) in PBS for 30 min prior to antibody staining. Fixed samples were analyzed with a TCS SP8 STED 3X system (Leica) in confocal mode using a 100 × 1.4 NA objective and HyD detectors. Images were acquired using the Leica LAS X software and converted to TIFF files using ImageJ (version 1.43; National Institutes of Health). Two-channel colocalization analysis was performed using the ImageJ software and the Pearson's correlation coefficient was calculated in a region of interest (ROI) using the 'Manders Coefficients' plugin developed at the Wright Cell Imaging Facility (Toronto, Canada). The ROI was determined as a mask created on the Golgi region formed upon Binary transformation (using the Default Thresholding given by the software) of the sum of both channels. Vesicle colocalization analysis was performed using ImageJ and was calculated using the 'Spots colocalization (ComDet)' plugin developed by Eugene A. Katrukha (Utrecht University, Netherlands). Golgi-derived tubules were quantified manually upon Z-projection (maximum intensity) performed using the ImageJ software.

## FLIP microscopy

Growth media of the HeLa cells seeded on Nunc Lab-Tek 8-wells 1.0 bottom glass chamber slides was supplemented with 25 mM 4-(2-hydroxyethyl)-1-piperazineethanesulfonic acid (HEPES) pH 7.4 prior to experiment. FLIP experiment was performed on a TCS STED CW system (Leica) in confocal mode using a ×63 NA objective and HyD detectors. Using the Live Data Mode tool of Las AF software (Leica), intermittent bleaching steps were combined with imaging of the whole cell during ~35 min.

When the FLIP experiments were performed on a GFP-containing protein, the series consisted of the following steps: (1) Image 512 × 512 pixels and pixel size ~75 nm, 200 Hz acquisition speed, 5% nominal power of an Argon Laser (488 nm). (2) Do 6× bleaching steps, keeping the same parameters as in step 1, but with a 100% laser power on a manually selected region encompassing the entire cell except the Golgi region. (3) Repeat step 1. (4) Pause for 40 s. (5) Repeat step 1. (6) Repeat steps 2–5 for 35 times.

When the FLIP experiments were performed on a mRFP-containing protein, the series consisted of the following steps: (1) Image 512 × 512 pixels and pixel size ~75 nm, 200 Hz acquisition speed, 15% nominal power of a HeNe Laser (543 nm). (2) Do 5× bleaching steps, keeping the same parameters as in step 1, but with a 100% laser power on a manually selected region encompassing the entire cell except the Golgi region. (3) Repeat step 1. (4) Pause for 42 s. (5) Repeat step 1. (6) Repeat steps 2–5 for 35 times.

To measure Golgi export rate of the protein of interest, the mean intensity of the region left by the photobleaching mask (Golgi-including region), $I(t)$, was measured over time and normalized to the initial intensity, $I_0 = I(t=0)$, using ImageJ and plotted using Prism 9.1.2 (GraphPad). Data were finally fitted to a one-phase decay curve fixing the 'plateau' to 0, that is $I(t)/I_0 = \exp(-t/\tau)$, from where the half-life, $t_{1/2} = \ln(2)\,\tau$, is extracted, and used to compare the Golgi residence time among different proteins.

## STORM imaging and data analysis

For STORM imaging, HeLa cells were cultured on 8-chambered glass bottom dishes (Thermo Scientific) at 37°C with 5% $CO_2$. Plasmids containing the sequence of the gene of interest were transfected and 24 hr after, cells were fixed with 4% paraformaldehyde in PBS at room temperature for 15 min. After fixation, cells were washed, permeabilized with 0.1% Triton X-100 (Fisher Scientific) (vol/vol) in PBS for 10 min, washed again, and blocked in 3% BSA (Sigma-Aldrich) (wt/vol) in PBS for 60 min. Then, cells were labeled with primary antibodies, as detailed in the 'Immunofluorescence microscopy' section. Donkey-anti-rabbit conjugated with AF405/AF647 (3:1) and donkey-anti-mouse conjugated with Cy3/AF647 (7:1) were used as secondary antibodies at 2 µg/ml final concentration.

STORM was performed in an imaging buffer containing 10 mM Cysteamine mercaptoethylamine (MEA) (Sigma-Aldrich), 0.5 mg/ml glucose oxidase (Sigma-Aldrich), 40 µg/ml catalase (Sigma-Aldrich), and 6.25% glucose (Sigma-Aldrich) in PBS (*Gómez-García et al., 2018*). Two-color images were acquired on a commercial microscope system (Nikon Eclipse Ti N-STORM system), from Nikon Instruments, equipped with a ×100 oil objective with NA 1.49 using oblique illumination. The detector was an ANDOR technology EMCCD iXon 897 camera, with a 256 × 256 pixel ROI and a pixel size of 160 nm. Fluorophore excitation was done using an Agilent technologies laser box with laser lines 405, 561, and 647 nm. NIS software (Nikon) was used for acquiring the data. In a pre-acquisition step the sample was irradiated with the 647 nm laser at 70% (~107 mW) power to bring a large majority of the dyes into a dark (off) state. STORM acquisition was then performed sequentially using 20 ms frames by maintaining the 647 nm laser at a constant power and using low power activator (405 or 561 nm) excitation (gradually increasing during acquisition from 0.5% to 100%). The acquisitions were stopped after ~70,000 frames.

STORM image reconstruction was performed in Insight3 software (*Huang et al., 2008*). The generated images were also corrected for cross-talked from images obtained from samples only labeled with 1 pair of antibodies. For the image reconstruction, a list containing the localization data for each channel were obtained. The degree of colocalization between both lists of localization was measured within a Golgi-containing ROI using the Coloc-Tesseler software (*Levet et al., 2019*). First, the software partition the image area into regions per localization where any given point within that region will be closer to that localization than to any other (Voronoi tessellation). Then, a first-rank density value, obtained from the area of the neighbor Voronoi areas, is assigned for each localization. Subsequently, these first-rank densities are represented in a logarithmic scaled scatterplot for each channel, always keeping channel A in x-axis and channel B in y-axis. The values in the scatterplot for channel A $(x^A, y^A)$ correspond to the first-rank density of a localization in channel A $(x^A)$ against the first-rank density of the localization of channel B in whose Voronoi area the point in channel A landed $(y^A)$. Vice versa for the scatterplot associated to channel B.

Finally, the Spearman's rank correlation coefficients are quantified as a measure of the strength and direction of the ranked variables plotted in each scatterplot. The Spearman's rank correlation coefficient is defined as $S^A = 1 - \frac{6\sum_i \left(r\left(x_i^A\right) - r\left(y_i^A\right)\right)^2}{n_A\left(n_A^2 - 1\right)}$, and $S^B = 1 - \frac{6\sum_i \left(r\left(x_i^B\right) - r\left(y_i^B\right)\right)^2}{n_B\left(n_B^2 - 1\right)}$, where $r\left(x_i^A\right)$ and $r\left(x_i^B\right)$ are the ranks of the first-rank density for channel A in the scatterplots A and B, respectively; $r\left(y_i^A\right)$ and $r\left(y_i^B\right)$ are the ranks of the first-rank density of the channel B for the scatterplots A and B, respectively; and $n_A$ and $n_B$ are the average number of neighbors per channel. As the Pearson's

correlation coefficient, the Spearman's rank correlation coefficient varies from 1 (perfect localization) to −1 (perfect antilocalization) and 0 representing a random distribution of localizations.

## Immunoprecipitation

HeLa cells were lysed with lysis buffer (0.5% Triton X-100 in PBS) containing protease inhibitors (Roche) for 15 min on ice, after which the samples were centrifuged at 16,000 × $g$ for 10 min at 4°C. The resulting supernatants were incubated for 16 hr while rotating at 4°C with 8 µg/ml anti-GFP antibody (Roche). Protein A/G PLUS-Agarose (Santa Cruz) was then added to the samples and incubated with rotation for 2 hr at 4°C. Immunoprecipitated fractions were washed two times with lysis buffer and two more times with PBS, and heated to 95°C for 5 min with ×1 Laemmli SDS sample buffer. The samples were subjected to SDS–PAGE and Western blotting with anti-GFP (Santa Cruz) and anti-RFP (Abcam) antibodies.

## Secretion assay

The growth media of HeLa WT and HeLa TGN46 KO cells seeded on a well of 6-well plate – and transfected with PAUF-Myc-His 48 hr previous to the experiment performance – was exchanged by pre-warmed (37°C) DMEM containing 1% penicillin/streptomycin and 2 mM L-glutamine. Cells were then incubated for 4 hr in an incubator at 37°C and 5% $CO_2$ to collect secreted PAUF-Myc-His protein. Then, media was collected in a 1.5-ml tube and centrifuged to avoid floating dead cells that may interfere. Protease inhibitor was added to sample and sample was concentrated down to 150 µl using Amicon Ultra-0.5 10 kDa Centrifugal Filter Units (Merck). On the other hand, cells were lydes on ice with 150 µl of ice-cold lysis buffer (1% SDS Tris–HCl pH 7.4). Samples were collected in a 1.5 ml tube using a scraper and centrifuged at maximum speed for 20 min at 4°C to remove cell pellet. Finally, 6× Laemmli SDS sample buffer was added to both, supernatant and cell samples, and incubated for 10 min at 95°C. The samples were subjected to SDS–PAGE and Western blotting with anti-Myc (X) and anti-TGN46 (AbD Serotec) antibodies. When cells were treated with BFA, the compound was added to the cells to a final concentration of 5 µg/ml 15 min before media was exchanged by FBS-deprived media and also to the collection media for the 4 hr incubation time of the secretion experiment.

## Statistics

Statistical significance was tested using one-way analysis of variance using GraphPad Prism 9.1.2, unless otherwise stated in the figure legend. Different datasets were considered to be statistically significant when the p-value was ≤0.05 (*); p-value ≤0.01 (**); p-value ≤0.001 (***); p-value ≤0.0001 (****).

## Acknowledgements

We thank members of the Single Molecule Biophotonics lab at ICFO, Chris Burd, Ivan Castello-Serrano, and Ilya Levental for valuable discussions. We thank Nathalie Brouwers and the Centre for Genomic Regulation/Universitat Pompeu Fabra FACS Unit (Barcelona) for help with cell sorting and the IRB Barcelona Advanced Digital Microscopy (ADM) Core Facility for help with microscopy. We also thank Jordi Andilla and Maria Marsal for technical support at the Super-resolution Light Nanoscopy (SLN) facility at ICFO. The authors acknowledge the Protein Technologies Unit, Centre for Genomic Regulation (CRG) (Barcelona, Spain) for providing the protein used in this study. CGC acknowledges a FPI fellowship awarded by MINECO in the 2018 call. PL, JVL, XS, MFGP, and FC acknowledge support from the Government of Spain (FIS2015-63550-R, BFU2015-73288-JIN, FIS2017-89560-R, RYC-2017-22227, PID2019-106232RB-I00/10.13039/501100011033/110198RB-I00, and PID2020-113068RB-I00/10.13039/501100011033; PID2022-138282NB-I00 project funded by the MCIN/AEI/10.13039/501100011033/FEDER, UE; and Severo Ochoa CEX2019-000910-S), Fundació Privada Cellex, Fundació Privada Mir-Puig, and Generalitat de Catalunya (CERCA, AGAUR), ERC Consolidator Grant CONCERT (GA 648201) and Advanced Grants NANO-MEMEC (GA 788546) and LaserLab 4 Europe (GA 654148).

## Additional information

### Competing interests

Xavier Salvatella: founder and scientific advisor of Nuage Therapeutics. Felix Campelo: Senior editor, *eLife*. The other authors declare that no competing interests exist.

### Funding

| Funder | Grant reference number | Author |
| --- | --- | --- |
| Ministerio de Ciencia e Innovación | FIS2015-63550-R | Maria F Garcia-Parajo |
| Ministerio de Ciencia e Innovación | BFU2015-73288-JIN | Felix Campelo |
| Ministerio de Ciencia e Innovación | FIS2017-89560-R | Maria F Garcia-Parajo |
| Ministerio de Ciencia e Innovación | RYC-2017-22227 | Felix Campelo |
| Agencia Estatal de Investigación | PID2019-106232RB-I00/10.13039/501100011033/110198RB-I00 | Carla Garcia-Cabau |
| Agencia Estatal de Investigación | PID2020-113068RB-I00/10.13039/501100011033 | Pablo Lujan Javier Vera Lillo Maria F Garcia-Parajo |
| Agencia Estatal de Investigación | PID2022-138282NB-I00 | Javier Vera Lillo Felix Campelo |
| Ministerio de Ciencia e Innovación | CEX2019-000910-S | Pablo Lujan Javier Vera Lillo Carmen Rodilla-Ramírez Maria F Garcia-Parajo Felix Campelo |
| European Research Council | CONCERT (GA 648201) | Carla Garcia-Cabau |
| European Research Council | NANO-MEMEC (GA 788546) | Pablo Lujan Javier Vera Lillo Maria F Garcia-Parajo Felix Campelo |
| Laserlab-Europe | GA 654148 | Maria F Garcia-Parajo Felix Campelo |

The funders had no role in study design, data collection, and interpretation, or the decision to submit the work for publication.

### Author contributions

Pablo Lujan, Conceptualization, Formal analysis, Investigation, Methodology, Writing - original draft, Writing – review and editing; Carla Garcia-Cabau, Formal analysis, Investigation, Methodology, Writing – review and editing; Yuichi Wakana, Formal analysis, Supervision, Investigation, Methodology, Writing – review and editing; Javier Vera Lillo, Hideaki Sugiura, Investigation, Methodology, Writing – review and editing; Carmen Rodilla-Ramírez, Software, Formal analysis, Investigation, Methodology, Writing – review and editing; Vivek Malhotra, Formal analysis, Funding acquisition, Writing – review and editing; Xavier Salvatella, Formal analysis, Supervision, Funding acquisition, Methodology, Writing – review and editing; Maria F Garcia-Parajo, Formal analysis, Supervision, Funding

acquisition, Writing – review and editing; Felix Campelo, Conceptualization, Formal analysis, Supervision, Funding acquisition, Investigation, Methodology, Writing - original draft, Project administration, Writing – review and editing

### Author ORCIDs
Carla Garcia-Cabau ⓘ https://orcid.org/0000-0003-0533-0642
Yuichi Wakana ⓘ http://orcid.org/0000-0001-7537-1293
Carmen Rodilla-Ramírez ⓘ http://orcid.org/0009-0007-9633-0686
Vivek Malhotra ⓘ http://orcid.org/0000-0001-6198-7943
Xavier Salvatella ⓘ https://orcid.org/0000-0002-8371-4185
Maria F Garcia-Parajo ⓘ http://orcid.org/0000-0001-6618-3944
Felix Campelo ⓘ http://orcid.org/0000-0002-0786-9548

Joint Public Review: https://doi.org/10.7554/eLife.91708.3.sa1
Author Response https://doi.org/10.7554/eLife.91708.3.sa2

## Additional files

### Supplementary files
• MDAR checklist

• Source data 1. Source data folder corresponding to the GraphPad files with the quantitative results shown in the figures, with the individual files named according to the respective figure panels they relate to.

### Data availability
*Source data 1* contains the numerical data used to generate the figures and figure supplements. *Figure 1—source data 1* contains the uncropped raw images of the blots shown in *Figure 1A*. *Figure 4—figure supplement 3—source data 1* contains the uncropped raw images of the blots shown in *Figure 4—figure supplement 3C*.

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
