## [Editor Report · eLife assessment]

This study provides the **fundamental** insight that TGN46, a single-pass membrane protein, acts as a cargo receptor for proteins at the Trans-Golgi Network. The authors demonstrate that the luminal domain of TGN46 is crucial for the incorporation of the soluble secretory protein PAUF into CARTS, a class of vesicles mediating TGN to surface traffic. The data presented are **compelling**, yielding a clear model for the sorting of cargos destined for secretion.

---

## [Referee Report · Joint Public Review]

TGN46 is a prominent TGN protein that cycles to the plasma membrane. It has been used as a TGN marker for many years, but its function has been unknown. This manuscript provides evidence that the luminal domain of TGN46 serves as a cargo receptor for incorporation of the soluble secretory protein PAUF into a class of TGN-derived carriers called CARTS. Interestingly, the luminal domain also plays an important role in the intracellular and intra-Golgi localization of TGN46, and it contains a positive signal for Golgi export in CARTS. They demonstrate that TGN46 loading into CARTS is not dependent on its cytosolic terminus using a deltaCT mutant. A speculative part of the manuscript proposes that the luminal domain of TGN46 might form biomolecular condensates that help to capture cargo proteins for export.

This is a very nice study that makes a significant contribution to the field. New insights are obtained regarding the function of TGN46 and the role of its various domains. Various potential interpretations of the data are presented in a balanced and constructive way.

---

## [Author Response]

The following is the authors’ response to the original reviews.

**Public Review:**
Lujan et al make a significant contribution to the field by elucidating the essential role of TGN46 in cargo sorting and soluble protein secretion. TGN46 is a prominent TGN protein that cycles to the plasma membrane and it has been used as a TGN marker for many years, but its function has been a fundamental mystery.In parallel, it remains unclear how most secreted proteins are targeted from the Golgi to the cell surface. These molecules do not contain conserved sequence motifs or post-translation modifications such as lysosomal hydrolases. Cargo receptors for these secreted proteins have remained elusive.Therefore, these investigations are likely to have a significant influence on the field.To gain an insight into the molecular role of TGN46 in sorting, they systematically test the impact of the luminal, transmembrane, and cytosolic domains. Importantly and against the current thinking, they demonstrate that the luminal domain of TGN facilitates sorting. Interestingly, neither the cytosolic nor the length of the transmembrane domain of TGN46 plays a role in cargo export. The effects of TGN46 depletion are specific as membrane- associated VSVG remains unaffected.Interestingly, TGN46 luminal domain also plays an important role in the intracellular and intra-Golgi localization of TGN46, and it contains a positive signal for Golgi export in CARTS. Rigorous, well-performed data support the experimental evidence.A speculative part of the manuscript, with some accompanying experimental data, proposes that the luminal domain of TGN46 forms biomolecular condensates that help to capture cargo proteins for export.One important point to discuss is that the effects of TGN46 KO are partial, suggesting that TGN46 stimulates the Golgi export of PAUF but is not essential for this process. The incomplete block is apparent in Fig 1 and in Fig 5D.

We thank the reviewers and the editorial team for their assessment and valuable feedback on our manuscript. Their supporting comments reinforce the significance of our findings.

Regarding the specific point raised about the partial effects observed in the TGN46 KO cell line, we acknowledge the importance of this issue, and we have addressed it in more detail in the revised version of our manuscript. The partial effects observed when using the TGN46 KO cell line are likely caused by several factors:

(1) It is important to consider the phenomenon of cell adaptation/compensation, which is documented to occur in gene knockout cell lines. Cells often respond to genetic perturbations by adapting to compensate the loss of a specific gene. These compensatory effects could potentially mitigate the full impact of TGN46 depletion and might explain the partial effects observed.

(2) Our data indicate that the absence of TGN46 reduces PAUF secretion, but does not completely block its export. These results align with our proposed role TGN46 in cargo sorting. In its absence, the secretory proteins likely exit the TGN via alternative routes/mechanisms, such as "bulk flow" or by entering other transport carriers in an uncontrolled manner. The partial redistribution of the TGN46-∆lum mutant into VSVG carriers (Figure 4D) supports this likelihood. Importantly, similar situations are observed when unrelated sorting factors are depleted from the Golgi membranes. For example, when the cofilin/SPCA1/Cab45 sorting pathway is genetically disrupted, the secretion of this pathway's clients is inhibited but not completely halted (e.g., von Blume et al. Dev.Cell 2011; J. Cell Biol. 2012).

(3) As suggested by the reviewers, it remains possible that TGN46 is not the sole player for cargo sorting. The existence of redundant or alternative mechanisms cannot be ruled out.

In our revised manuscript, we have now provided a more in-depth discussion of these factors and their potential contributions to the observed partial effects in TGN46 KO cells (lines447-463). We believe that a comprehensive exploration of these possibilities will improve our understanding of the role(s) of TGN46 in cargo sorting and TGN export.

Recommendations for the authors: please note that you control which revisions to undertake from the public reviews and recommendations for the authors

The reviewers were unanimously enthusiastic about your work. They felt that the manuscript could be significantly improved mostly through careful re-wording, additional explanations and some figure modifications.

We thank the reviewers and the editorial team for their enthusiastic assessment of our findings. Their positive feedback is reassuring.

We have now addressed the reviewers' suggestions to improve the clarity of our manuscript. Specifically, we have improved various aspects of the text that may have lacked clarity in the initial submission. This includes a thorough re-writing of respective sections to ensure that the content is more accessible and reader-friendly (see detailed answers to the additional points below). Furthermore, we have carefully followed the recommendations related to figure modifications.

Please mention the species (human) in the title.

We have changed the title according to the suggestion. The revised title now is: "Sorting of secretory proteins at the trans-Golgi network by human TGN46". In addition, we have also added the word "human" in the abstract ("... we identified the human transmembrane protein TGN46 as a receptor for the export of secretory cargo protein PAUF in CARTS ...").

Additional points:The main Figures only show quantifications that are challenging to understand without fluorescence micrographs. We suggest putting the micrographs of the fluorescence images (Figures S2A and B) into the main Figure 2 (before 2B and 2C)-the same in Figures 3 and 4.

Following the reviewers' suggestion, we have incorporated the fluorescence micrographs (included as figure supplements in the initial submission) into the main figures 2–5. Given that these additions have introduced a significant number of extra figure panels, we have carefully re-designed the figure layout to accommodate all the necessary information. This has involved that the FLIP data from old Figs. 2–4 is now included as a new Fig. 3; and the split of old Fig. 4 in the new Figs. 5,6. The supporting figures have also been rearranged accordingly. In addition, we have changed the color palette of the micrographs, in which now the dual-color images are presented in color-blind-friendly green and magenta, instead of green and red as previously. We believe that in this revised manuscript, all data and micrographs are clearly presented.

For figures such as Fig. 1B, the mean and SD positions are hard to see for the data plotted as solid black dots. Maybe hollow circles would be better.

The reviewers are right and we apologize for any difficulty in discerning the mean and SD positions from the figure. In our revised version, we have made the necessary modifications to all the figures where data points were plotted as solid black circles by converting them into empty black circles, as suggested by the reviewers.

In the right side of Fig. 1A, is the difference in PAUF secretion between WT and KO cells truly significant? The meaning of the number of asterisks should be given in the legend. Only one asterisk is shown, suggesting that the significance is low.

In our revised manuscript, we have included comprehensive information about the statistical significance, such as the statistical test used, p-values/asterisk meaning, and any other relevant details. In addition, we have included the lines connecting the individual data points corresponding to the different replicates of the secretion assays (WT vs KO).

Experiments such as the one in Fig. 1C may be better described as iFRAP rather than FLIP.

We appreciate the reviewers' attention to the experimental methods used, e.g., in Figure 1C. We actually performed FLIP experiments rather than iFRAP, and we acknowledge that this might not have been stated clearly in our initial submission. The distinction between iFRAP and FLIP lies in the frequency of photobleaching. In iFRAP, photobleaching occurs only once at the beginning of the experiment, whereas FLIP involves repeated photobleaching (FLIP is sometimes also referred to as "repeated iFRAP"), which was conducted in our experiments. Specifically, in our experiments we performed repeated photobleaching at a relatively slow rate (approximately once per minute; every two imaging frames).

We understand the potential source of confusion, which may have arisen from the references we provided to introduce our FLIP experiments (Hirschberg et al. 1998; Patterson et al.2008). In those papers, almost all results were obtained using iFRAP and not FLIP. In light of this feedback, we have made significant efforts in our revised manuscript to clarify the terminology and procedure used in our experiments (lines 148-154). These revisions have improved the understanding of our findings and we appreciate the reviewers' suggestions.

When using iFRAP to measure the Golgi residence time of a TGN46 construct that has a cytosolic tail, shouldn't recycling from the plasma membrane be taken into account? Unlike a secreted protein, TGN46 will never show complete loss of signal from the Golgi.

The reviewers are right: for a TGN46 construct that can efficiently recycle back to the TGN from the cell surface, an iFRAP experiment would not report solely the protein residence time at the Golgi. We concur with the reviewers, and we'd like to clarify that the reason we performed FLIP experiments, as opposed to iFRAP, was precisely to address this concern. In an iFRAP experiment, where photobleaching occurs only once at the beginning, the fluorescence decay within the Golgi area would indeed consist of two components: a decay due to the export of the protein and an increase in fluorescence due to the protein that had been exported (after the initial photobleaching) and then recycled back to the Golgi area. In contrast, our choice of conducting FLIP experiments, with repeated photobleaching of the pool of fluorescent protein outside the Golgi area (approximately once per minute), minimizes the influence of recycling. Consequently, the loss of fluorescence in the Golgi area in our FLIP experiments predominantly reflects the protein's export. We acknowledge that this distinction was not adequately communicated in our initial submission and we have emphasized these points in the revised version of the manuscript (lines 230-234).

Lines 274 to 285 are confusing and controversial. The author argues that the transmembrane domain does not impact TGN localisation and cargo packaging. Later, they state, "These data further support the idea that the slower Golgi export rate of TGN46 mutants with short TMDs is a consequence of their compromised selective sorting into CARTS".

We appreciate the reviewers' attention to the potential confusion regarding the impact of the TMD on TGN localization and cargo packaging. Actually, our results indicate that the length of the TMD does not seem to have an impact in intra-Golgi protein localization (Fig. 4B,C) but they do play a role in incorporation into CARTS (Fig. 4D,E). We have now clarified this in the text (lines 283-284; 296-297).

That being said, these results were also surprising to us initially. However, upon closer examination of the amino acid sequence of the cytosolic domain of TGN46, we noticed a possible side effect of shortening its TMD. Shortening the TMD of TGN46 could lead to the partial burial of highly charged residues from TGN46 cytosolic tail (HHNKRK...) into the membrane, potentially affecting its behavior. For that reason, we constructed the TGN46 ∆cyt ST-TMD mutant, which features a short TMD (ST TMD) and lacks the potential interference from the cytosolic tail (see also lines 307-320). Notably, this mutant showed a phenotype similar to that of TGN46-Δcyt, and to that of full length TGN46, particularly in terms of intra-Golgi localization and CARTS specificity. We acknowledge that the interpretation of these results can be debated, and we have ensured that the revised manuscript captures these nuances. Additionally, we have realized that the organization and presentation of these results may have caused confusion, particularly concerning the placement of the results from the GFP-TGN46 ∆cyt ST-TMD mutant. To address this, we have reorganized old Figures 2 and 3 to ensure that the results of the GFP-TGN46 ∆cyt ST- TMD mutant are presented with the short TMD mutants. These adjustments have greatly improved the overall flow of our manuscript. We thank the reviewers for their valuable feedback.

In lines 444-446 in the Discussion the argumentation is confusing. The experiment shows that the cytosolic domain of TGN46 has no impact on TGN46 localisation or cargo packaging into a nascent vesicle. At the same time, the authors mention that a cytosolic complex composed of Rab6 and p62 is required to generate CARTS.

We are grateful for the reviewers' feedback regarding our argumentation in lines 444-446. Indeed, our results indicate that the cytosolic tail of TGN46 does not play a major role in packaging of TGN46 in CARTS and in PAUF secretion. However, it is important to acknowledge that our findings do not rule out the possibility that TGN46 might have a dual function at the TGN. It could potentially play a role in mediating or controlling the export of other cargo proteins by alternative mechanisms/routes, which could, in part, depend on its cytosolic domain.

This complexity is consistent with the open question regarding the role of the cytosolic Rab6- p62 complex in CARTS biogenesis. Interestingly, in experiments reported in Jones et al. (1993), a Golgi budding assay was used to test the involvement of the cytosolic domains of TGN38 and TGN41 in budding of Golgi-derived carriers that contain the transmembrane cargo protein pIgA-R (polymeric IgA-receptor). The authors showed that the budding of these carriers was blocked upon incubation of the Golgi membranes with peptides against the cytosolic tail of TGN38/41 but not peptides against their lumenal domain. However, in the latter experiment, they used a peptide formed by the 15 N-terminal residues of TGN46, which might not functionally block the entire lumenal domain (>400 residues). Our results with reference to earlier results in the field will serve as a basis for further exploring the role(s) of TGN46 in cargo export beyond the scope of the present study.

In summary, these are all very important points (we thank again the reviewers for highlighting them), which we have now carefully addressed in the revised version of our manuscript (lines 476-485).

The phase separation experiments are exciting. However, they are not necessary. They may be more confusing than helpful for the following reasons:• The authors use very high protein concentrations and crowding reagents. Any protein would condense under these conditions.The protein was produced in bacteria so that it won't have post-translational modifications, especially glycosylation, possibly the most critical drivers of phase separation.There was no test of direct binding of PAUF with TGN46

We appreciate that the reviewers share our excitement about our preliminary phase separation experiments. Likewise, while we initially included these experiments in the "Ideas and speculation" section due to their exciting nature, we concur with the reviewers that their preliminary nature and the experimental conditions used to obtain them raise valid concerns.

In light of these considerations and to prevent any potential confusion for the readers, we have decided to follow the advice of the reviewers. We have removed the phase separation experiments and data from the revised manuscript. Instead, we have retained a simplified and concise "Ideas and speculation" section, in which we propose condensate formation as a potential mechanism by which TGN46 functions as a cargo sorter at the TGN (lines 580- 620).

The authors reference S5A as the localisation between TGN46deltaLUM images, however, we believe they are referring to fig. S7.

We apologize for the oversight in referencing the figure and thank the reviewers for bringing this to our attention. We have amended this in the revised version.

The authors write "remarkably, the amino acid sequence of rat TGN38 is largely conserved amongst other species, including humans (>80% amino acid identity between rat TGN38 and human TGN46)". To understand if this is remarkable, the authors should use the average identity between rat and human proteins.

We are grateful for the reviewer's insightful comment. Indeed, as the reviewer hints, the average identity between the rat and human proteomes is of the same order of magnitude as the identity reported between rat TGN38 and human TGN46. We therefore acknowledge that the term "remarkable" may not be suitable in this context and could lead to potential misinterpretation. In the revised version, we have removed the term "remarkably".